# Active topological glass

Jan Smrek [1,2,3]*, Iurii Chubak [1,3], Christos N. Likos [1,4] & Kurt Kremer [2,4]

The glass transition in soft matter systems is generally triggered by an increase in packing fraction or a decrease in temperature. It has been conjectured that the internal topology of the constituent particles, such as polymers, can cause glassiness too. However, the conjecture relies on immobilizing a fraction of the particles and is therefore difficult to fulfill experimentally. Here we show that in dense solutions of circular polymers containing (active) segments of increased mobility, the interplay of the activity and the topology of the polymers generates an unprecedented glassy state of matter. The active isotropic driving enhances mutual ring threading to the extent that the rings can relax only in a cooperative way, which dramatically increases relaxation times. Moreover, the observed phenomena feature similarities with the conformation and dynamics of the DNA fibre in living nuclei of higher eukaryotes.

[1] Faculty of Physics, University of Vienna, Boltzmanngasse 5, A-1090 Vienna, Austria. [2] Max Planck Institute for Polymer Research, Ackermannweg 10, 55128 Mainz, Germany. [3] These authors contributed equally: Jan Smrek, Iurii Chubak. [4] These authors jointly supervised this work: Christos N. Likos, Kurt Kremer. *email: jan.smrek@univie.ac.at

Tremendous interest has been devoted to understanding of the glass transition driven by an increase in packing fraction or a decrease in temperature in soft and deformable systems[1,2]. Ultra-soft particles can be realized experimentally using polymers, such as long polymeric stars or rings, which are highly deformable but possess a fixed topology imposed during the synthesis. While it is known that the polymer topology has strong impact on the stress relaxation mechanism[3], it has been questioned if the topology can independently induce a glass transition[4].

Such a topological glass has been conjectured for the system of long, unknotted, and nonconcatenated polymer rings in dense equilibrium solutions[4,5]. The rings cannot cross each other, but are known to thread—one ring piercing through an eye of another ring, which temporarily topologically constrains the motion of the two rings. Mutual threadings of many rings can yield a conformation that is relaxed only if a cascade of threadings is sequentially undone, which could give rise to very long relaxation times, e.g., exponential in the ring length[6]. The glassy behavior in an equilibrium melt of rings has been observed in computer experiments, but only under a pinning perturbation, which immobilizes a fraction $f$ of all rings. Then, for sufficiently long rings and high density a glassy behavior can be extrapolated to $f \to 0$[4,5]. Unfortunately, the conjectured critical length, 90 entanglement lengths, of unpinned rings is currently beyond the reach of experiments and, although the pinning deepens our theoretical understanding of the glass transition[7–9], creating it experimentally to drive the topological glass transition would be challenging.

Whereas many questions remain open in the traditional glass transition of passive Brownian particles, recently a whole new research direction has been opened by considering system composed of active particles that are driven by non-thermal fluctuations[10]. While, intuitively, activity opposes glassiness by enhancing mobility of the particles, some active models can exhibit a more complex behavior as a function of the active control parameters. For example, increasing the persistence time of the active Ornstein-Uhlenbeck particles can either glassify or fluidize the active system, depending on the particular state point, as a result of nontrivial velocity correlations in the system[11–14]. Indeed, some system properties, such as the time-dependent effective temperature are pertinent to active fluids and render also the corresponding glass transition distinct from the passive one. In particular, the location and the existence of the glass transition of active fluids are dependent on the microscopic details of the activity mechanism. Nevertheless, close to the transition region, universal features of the passive glassy dynamics have been found recently for active spin-glasses[10] and self-propelled particles in the non-equilibrium mode-coupling theory (MCT)[14]. For instance, the scaling of the relaxation time with activity control parameters is governed by the same exponent as in the passive MCT.

The impact of topology on the active glassy states has been studied almost solely in the context of active particles confined to a topologically nontrivial space, in particular a sphere[15–19]. There, a range of dynamic phenomena arises as a consequence of the competition between directed flows, characteristic for active matter in euclidean space, and the 'hedgehog theorem'[20] that asserts the existence of topological defects in a smooth vectorial field on a sphere. In contrast to these studies, where the topology is a global property of space, here we focus on a system where the active particles themselves are not point-like but feature intrinsic circular topology, the embedding space being Euclidean with periodic boundary conditions.

Glassy dynamics also appears in various biological contexts ranging from the bacterial cytoplasm[21] to collective cell migration in tissues[22]. While on the subcellular level, glass-like properties have been attributed to the high crowding, as well as size and interaction heterogeneity of the constituents[21,23], the confluent tissues modeled using vertex models exhibit a new type of rigidity transition at constant density without[24] and with active motion[25], attributed to an interplay of cells' shape and persistence of motion. Although there is no topology involved in these models and, therefore, they are inherently different from our present work, the transition occurs due to shape changes at constant density similar to the case studied here.

For dense solutions of ring polymers, we show that making the rings locally more mobile by introducing a moderate segmental activity, the system reaches a glassy state with dramatically slowed-down relaxation. This novel state of matter, the 'active topological glass', is a consequence of the interplay between internal polymer topology, activity, and the crowded polymer environment. In contrast to the conjectured equilibrium topological glass, here no imposed pinning is necessary and only relatively short rings are sufficient to observe the transition. Moreover, contrary to well-studied polymer glasses[26], where the monomers are arrested due to their nearest neighbors, here the centers of mass of the whole chains are inhibited due to multibody, long-range effects of topological constraints. After detailed account of the physics of the active topological glass, we discuss its relevance for the organization and dynamics of chromosomes in living eukaryotic cells.

## Results

**Dynamics after the onset of activity.** We start with a large, well-equilibrated, concentrated solution of $M = 1600$ passive, uncrossable, unknotted, and nonconcatenated rings of length $N = 400$ using the well-established polymer model as in Halverson et al.[27,28]. A consecutive segment of length $N_h = 50$ monomers is made active on each chain by subjecting it to thermal-like fluctuations of temperature three times higher than the rest of the chains (see Methods for model details).

After switching on the activity, the initial equilibrium uniform spatial distribution of the active (hot) and the passive (cold) segments alters (Fig. 1a). They progressively segregate to compensate for the local pressure imbalance and to decrease entropy production[29–31]. Simultaneously, we observe a gradual but dramatic conformational change of the rings as revealed by the mean radius of gyration $\langle R_g \rangle$ and other shape parameters (see Supplementary Note 1) that comes to a standstill after several equilibrium diffusion times $\tau_{\mathrm{diff}} \simeq 4 \times 10^5 \tau$ (see Methods for the definition). In an equilibrium system of linear block copolymers, the colocalization of like-blocks can drive local density inhomogeneities that affect chain conformations. We show in Supplementary Note 3 that analogous effect is not responsible for the conformational changes in the present non-equilibrium system. Foremost, we do this by simulating a system with a low fraction of active chains where the colocalization of hot segments is not present but the conformational changes persist (Supplementary Fig. 9). In addition, we show on a simplified effective equilibrium model that in a fully phase separated system the ring conformations do not differ substantially from the homogeneous equilibrium case (Supplementary Fig. 4).

We observe that the hot segment is usually localized at one of the ends of a tree-like, doubly folded conformation (Fig. 1b). This suggests that the conformational change is caused by the differences between the dynamics of the active and passive segments. As the active segment undergoes stronger thermal fluctuations, its diffusivity is enhanced in comparison to the passive one. In fact, it essentially drags the cold tail through the mesh of other chains (Supplementary Movie 1). We illustrate

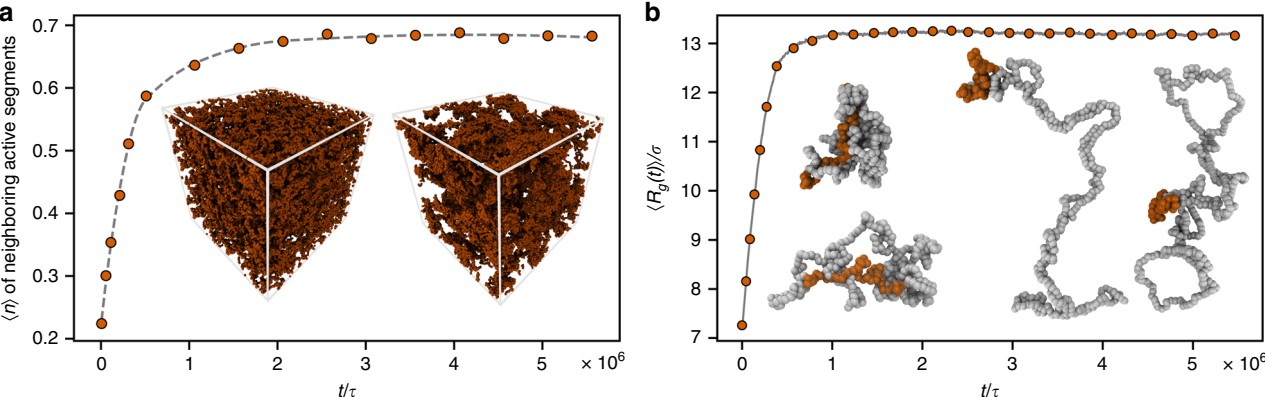

**Fig. 1 Structural properties of the system as a function of time after the onset of activity. a** Mean number of neighboring hot segment as a function of time. Two hot segments are neighboring if their centers of mass are within the distance of their radius of gyration ($3.25\sigma$). A completely uniform distribution corresponds to a value 0.3. The dashed gray line is a guide to the eye. Insets: snapshots of the system showing only the hot segments at an early (left) and a late (right) time. **b** The mean radius of gyration $\langle R_g \rangle$ obtained as an average over all rings at a given time $t$ after the onset of activity (Eq. (4) in Methods). Insets: snapshots of two rings in equilibrium (left) and two at late times (right). The hot segment on the active rings is shown in orange (on the equilibrium rings the orange segment is highlighted for comparison only and has the same temperature as the rest of the system).

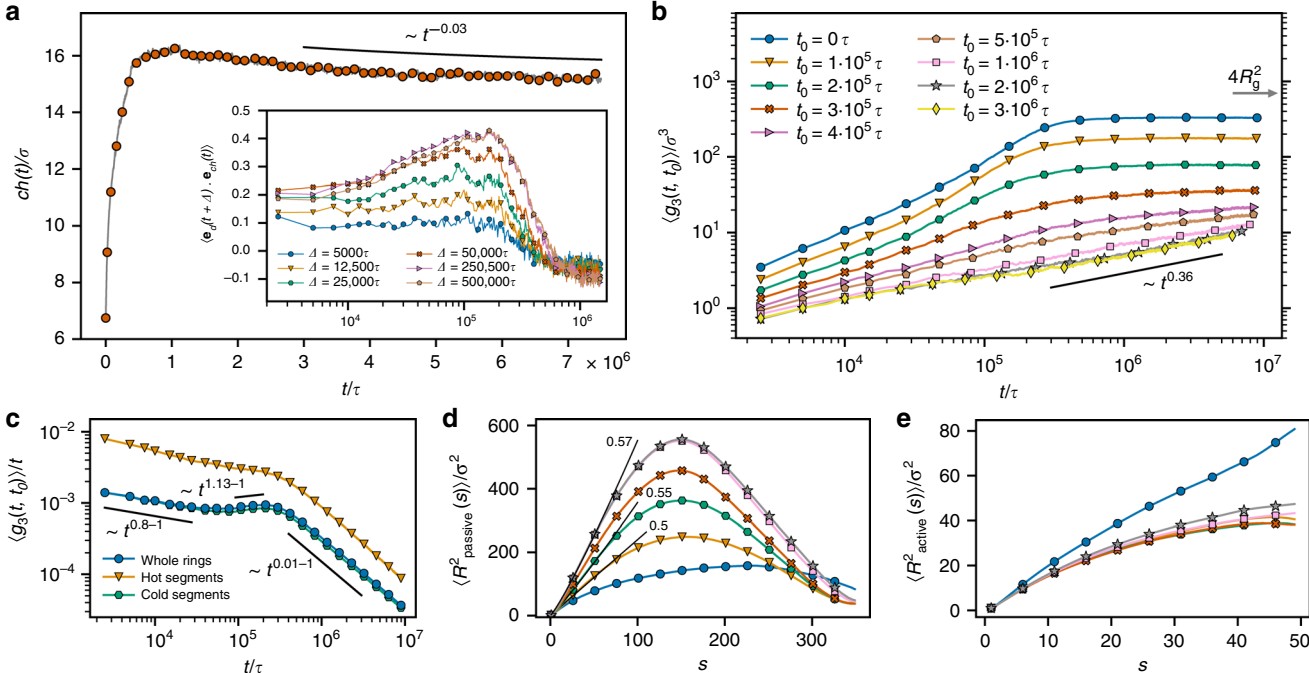

**Fig. 2 Evolution of the dynamics. a** Mean magnitude of the cold–hot vector **ch** as a function of time. Inset: correlation of the **ch** direction with the direction of the center of mass displacements of the rings in different lag times $\Delta$ as a function of time $t$ after the activity onset ($\mathbf{e}_d(t + \Delta)$ and $\mathbf{e}_{ch}(t)$ are unit vectors in the direction of $\mathbf{d}(t) = \mathbf{R}(t + \Delta) - \mathbf{R}(t)$ and $\mathbf{ch}(t)$, respectively). **b** Mean squared displacement of the centers of mass of the rings, $\langle g_3(t, t_0) \rangle$, as a function of time $t$ measured from different times $t_0$ after the activity onset. **c** $\langle g_3(t, t_0) \rangle / t$ as a function of time for $t_0 = 0$. **d, e** Time-resolved mean squared internal distance for the passive (**d**) and the active (**e**) segment $\langle R^2(s)_{\text{passive/active}} \rangle$, computed for each segment length $s$ as the squared distance of the endpoints of the segment averaged over the segments position within the passive/active block of a ring and averaged over rings. The black straight lines in (**d**) emphasize the scaling behavior $\langle R^2(s) \rangle \sim s^{2\nu_{\text{trail}}}$ for low $s$ with the numbers indicating $\nu_{\text{trail}}$ for a few characteristic times after the activity onset. Panels **b, d**, and **e** share the same legend shown in (**b**).

such emergent directionality of the ring's motion by computing the 'cold-hot' vector **ch** connecting the centers of mass of the cold and hot segments of each ring. After the onset of the activity, the mean magnitude of **ch** initially grows (Fig. 2a). During this time, the direction of the motion is correlated with the direction of **ch** (Inset of Fig. 2a). At later times, |**ch**| decreases very slowly, generating a weak anticorrelation with the displacement vector, which is connected to the strengthening of topological constraints

as detailed later. As we show in Supplementary Note 2, systems without topological constraints lose their directionality on a microscopic timescale, but the dense polymer environment generates a more intriguing global dynamics.

To describe the dynamics, we track in time the mean squared displacements of the rings centers of mass $\langle g_3(t, t_0) \rangle$ (see Eq. (5) in Methods and note that the mean is taken over the rings only because the dynamics is not stationary in general). Figure 2b

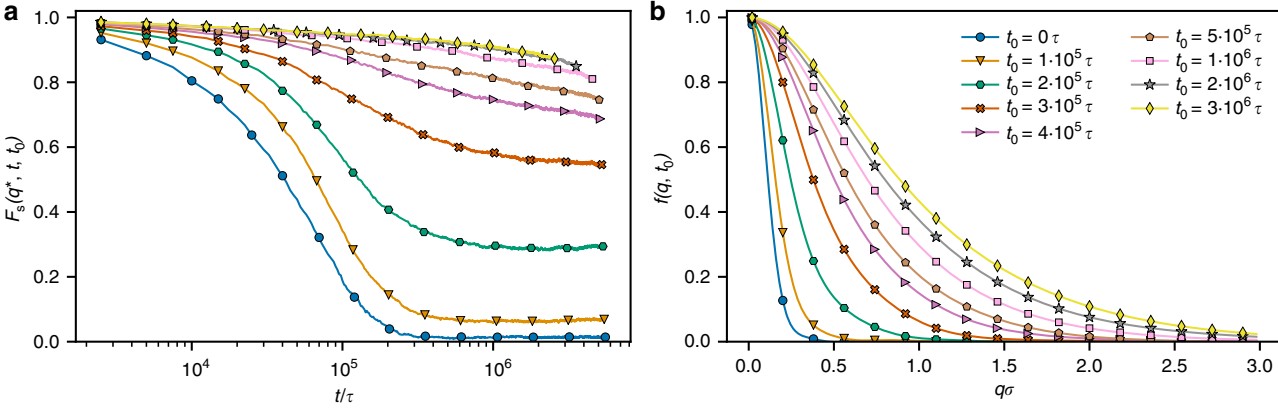

**Fig. 3 Evolution of the relaxation. a** Self-intermediate scattering function $F_s(q^*, t, t_0)$ as a function of $t$ for different $t_0$ at $q^* \sigma = 0.35$. **b** Non-ergodicity parameter $f(q, t_0)$ defined as $f(q, t_0) = F_s(q^*, t_{max}, t_0)$, where $t_{max}$ is total time of our simulation, as a function of $q$ for different times $t_0$ after the activity onset. Panels **a** and **b** share the same legend shown in (**b**).

shows $\langle g_3(t, t_0)\rangle$ as a function of time $t$ measured from various times $t_0$ after the activity onset. $\langle g_3(t, 0)\rangle$ increases initially and after $10^6\tau$ displays a dramatic slow-down. The crossover time is the same as for the structural changes which underlines the fact that the two effects are dependent. More insight is provided by Fig. 2c, where $\langle g_3(t, t_0)\rangle/t$ is plotted as a function of time separately for the active and the inactive segment. The initial decrease corresponds to subdiffusive $\langle g_3(t, t_0)\rangle \sim t^{0.8}$ regime consistent with the equilibrium rings dynamics below the diffusion time[28]. The following regime shows again the directional dynamics of the rings and explains its origin. While the passive segment, and hence the whole center of mass, move superdiffusively (exponent $1.13 \pm 0.01$), the active one temporarily displays standard diffusion (exponent 1). The both stages last for less than a decade in time before both segments cross over to an arrested state with exponent very close to zero.

The superdiffusive regime is a consequence of the specific non-equilibrium dynamics through a mesh of topological constraints. As the detailed balance is violated due to the coupling to different thermostats, the pulling of the active segment forwards is stronger than the pulling of the cold tail backwards. The hot segment robustly explores new sites that are spontaneously freed due to density fluctuations and progressively drags behind itself the cold tail (see Supplementary Movie 1). Furthermore, such motion of the active segment away from its cold tail through the environment of neighboring rings introduces new topological constraints that the cold tail must obey. These constraints restrict the transversal motion of the chain. Finally, when the chain is getting more stretched some time after the activity onset, the motion of the hot segment backwards to the cold tail is compromised by the chain flexibility and, therefore, the motion away from the tail prevails. As a result, the cold tail follows the hot head slowly, but ballistically along a trail imposed by the topologically constrained neighboring rings. At these length scales, the trail is characterized by the size $R$ of the static conformation of the tail and it scales with the contour distance $s$ as $R(s) \sim s^{\nu_{trail}}$ where the exponent $\nu_{trail} = 0.57 \pm 0.01$ (Fig. 2d). Therefore, the directed dynamics ($s \sim t$) along such contour is superdiffusive with $\langle g_3(t, t_0)\rangle \sim R^2(s(t)) \sim t^{2\nu_{trail}}$, which is in agreement with our observation (Fig. 2c). Moreover, the onset time of the superdiffusion $t \approx 10^5\tau$ is consistent with the onset time of a configuration that is more open than a random walk, that is $\nu_{trail} > 0.5$ as seen in Fig. 2d, and the dragging mechanism is consistent with the fact that the endpoints of the active segment are closer to each other than in the equilibrium case (Fig. 2e).

**Glassy behavior**. Subsequently, the system is slowing down, as revealed by $\langle g_3(t, t_0)\rangle$ (Fig. 2b). To characterize the slowing down of the relaxation of the rings in more detail, we measured the self-part of the intermediate scattering function (ISF) $F_s(q, t, t_0)$ (Eq. (6) in Methods). As shown in Fig. 3a, $F_s(q^*, t)$ depends on $t_0$, similar to the aging in classical glasses[32]. After about $2 \times 10^5\tau \simeq \tau_{diff}$ the system fails to relax and $F_s(q^*, t)$ plateaus at a non-vanishing value, defined as the non-ergodicity parameter $f_q$ (Fig. 3b). A striking characteristic of the ensuing arrested state is that it features a single, $\beta$-relaxation process and the subsequent $\alpha$-relaxation is absent, in contrast to the common, two-step relaxation scenario[33] encountered for polymer glasses[34] or for repulsive colloids[35]. Indeed, we have not been able to observe the $\alpha$-relaxation, despite the fact that we have extended our simulations to very long times, over $22\tau_{diff}$. The absence of a two-step process is a feature associated with continuous, type-A glass transitions, as opposed to the discontinuous, type-B transitions[36–38], and it implies the presence of higher-order singularities, the so-called $A_3$- and $A_4$-critical points, in the framework of MCT[39–42]. There are strong indications that the system at hand features such higher-order singularities, a point to which we will return in the Discussion section. At later measurement start times, the non-ergodicity parameter is higher, that is, more wave vectors fail to relax, and we cannot even observe the $\beta$-relaxation (more details in Supplementary Note 4). In fact, as the system evolves toward a steady state ($10^6\tau < t < 2\times 10^6\tau$), progressive strengthening of the topological constraints in the system takes place, which restricts the rings' motion and leads to the rise of the plateau height. This characteristic is similar to the strengthening of glassy behavior for colloidal systems as the packing fraction grows[43].

After about $2 \times 10^6\tau$, $\langle g_3(t, t_0)\rangle$ becomes independent of $t_0$ and only small changes in $\langle g_3(t)\rangle$ and $F_s(q^*, t)$ are noticeable. These are due to the local system explorations of the hot segments (Supplementary Fig. 2), reminiscent of a confined diffusion with occasional constraint release. $\langle g_3(t)\rangle$ is strongly subdiffusive ($\sim t^{0.36}$) and very slow, typical for polymeric glasses[26]. The corresponding relaxation time, extrapolated as $\langle g_3(\tau_{relax})\rangle = 4R_g^2$, is $\tau_{relax} \approx 10^{12}\tau$, being more than six orders of magnitude higher than the equilibrium one. Such a strong dependence of the relaxation time on the control parameter is a hallmark of glassy systems. Linear polymers in an equilibrium melt relax slower than rings due to a reptation relaxation mechanism, but for the lengths considered here, their relaxation time is only about twice as large as that of the equilibrium rings[28]. Therefore, the glassy behavior

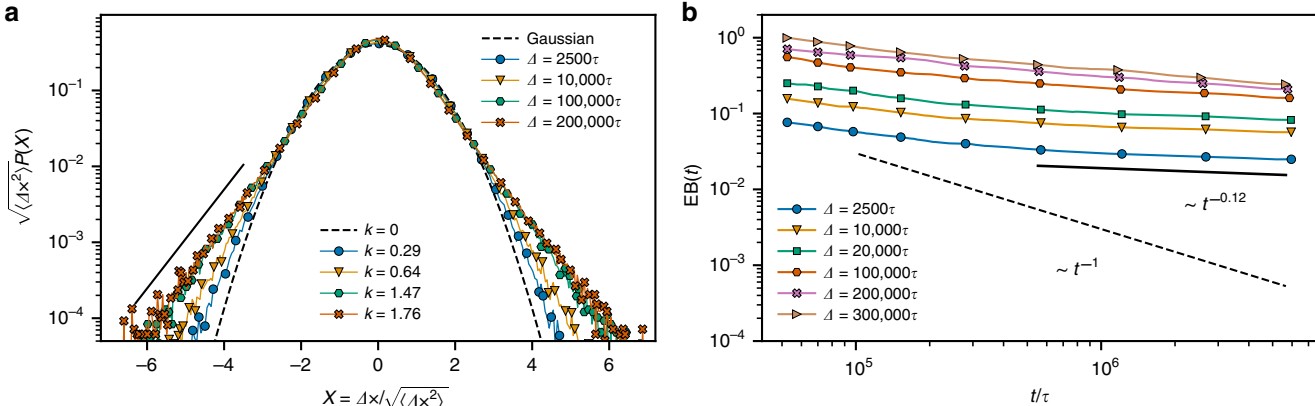

**Fig. 4 Properties of the steady state. a** Normalized distributions of the center of mass displacements in the *x*-direction for different lag times Δ measured after $2 \times 10^6 \tau$. Identical distributions are found for other directions (not shown). The bottom legend contains the computed values of the kurtosis $\kappa = (m_4/m_2^2) - 3$, where $m_i$ is the *i*-th moment of the distribution, indicating strong non-Gaussian character of the distributions upon increasing Δ. **b** Time evolution of the ergodicity breaking parameter (7) for various lag times Δ.

of partly active rings, although they are somewhat doubly folded and hence remind of linear polymers of length $N/2$, cannot be attributed to the reptation-like relaxation.

To further support the evidence of glassy dynamics, we measure the probability distribution of $1d$ displacements for various lag times Δ in the glassy regime, that is for $t > 2 \times 10^6 \tau$ (Fig. 4a). For short $\Delta = 2500\tau$, the distribution is close to Gaussian, characterizing the standard diffusion (also in the non-equilibrium two-temperature case[44]), whereas for longer lag times ($\Delta > 10^5 \tau$) it becomes markedly non-Gaussian (see also Supplementary Fig. 5a). Tiny displacements and their non-Gaussian distribution characterize a local constraint (cage) exploration which is another hallmark of glassy systems[45]. Interestingly, for long lag times the tails of the distribution are just simply exponential as in the equilibrium topological glass induced by pinning perturbations[5]. We attribute these 'fat' tails to a constraint release and a short relocation of the hot segments of an individual chain (Supplementary Movie 1). In addition, we measured the ergodicity breaking parameter EB[5], defined by Eq. (7) in Methods. The EB characterizes how quickly (averaged over rings), the time average of a single ring $g_3$ converges to the ensemble average $\langle g_3(\Delta) \rangle$. While in equilibrium EB typically decays as $t^{-1}$, the constraints with diverging lifetimes make glassy systems nonergodic with EB $\sim t^0$. We plot EB in the steady state regime (Fig. 4b), where we see a dramatic slowing down with exponent around $-0.1$ even for short lag times (see also Supplementary Fig. 5b for $g_3(t, t_0, \Delta)$ of individual rings as a function of the integration time).

**Threading analysis**. We now show that the mutual ring threadings are responsible for the glassiness. We analyzed the ring threadings using computationally spanned minimal surfaces on the ring contours (Fig. 5a). An intersection of one ring's contour through another ring's minimal surface defines a threading of the second ring. This method was used to clarify the extent and the role of threadings in equilibrium ring melts[46,47] (details in Methods).

We characterize the threading depth in terms of the so-called separation length $L_{sep}$, defined by Eq. (9) in Methods. It approximates how much of the threading rings material is on one side of the threaded ring. Then, the ratio $Q = L_{sep}/(N - L_{sep})$ defines the relative portion of the material on one side compared with the other side of the threaded ring's surface. This ratio provides a model-independent view on threadings because

its distribution in equilibrium is insensitive to the polymer model above the entanglement length scale $N_e$[47]. We found that more ring pairs are involved in threadings and that they are progressively deeper compared with equilibrium (Fig. 5c, d, respectively). Moreover, we found a positive correlation of the location of threadings with the local mechanical stress (Supplementary Fig. 3).

The glassy behavior should be connected to the emergence of a system-spanning cluster of rings that fail to relax due to mutual threadings. We define two rings belonging to the same cluster if at least one of them threads the other one with depth $L_{sep} \geq L_{cutoff}$. Therefore, we discriminate the cluster structure by the depth of threadings. Figure 5b shows the relative size of the biggest cluster as a function of $L_{cutoff}$ for different times after the activity onset. For low $L_{cutoff}$, the whole system is one cluster as each ring has shallow threadings with many of its neighbors. However, in equilibrium ($t_0 = 0$) the threadings are rarely deeper than $3N_e$ and therefore for $L_{cutoff} > 100 \simeq 3N_e$ the system can be viewed as a set of many small disconnected clusters. This sharply contrasts with the structure in the glassy regime ($t_0 > 10^6 \tau$), where all rings form a single cluster practically independent of the choice of $L_{cutoff}$ as a significant number of threadings of any depth occurs (Fig. 5b).

While the cluster profile remains stable in the glassy regime, we still observe threading and unthreading events between ring pairs. Numbers of these events, however, balance each other, resulting in a steady state (Fig. 6a). Interestingly, the threading lifetime distribution shows a power-law with a peak at late times, meaning that the majority of threadings are persistent and survive for the total duration of our simulations with a minority having a short lifetime. The bimodal character is likely a consequence of the fat power-law tail as revealed by the shape of the distribution measured at different times after the activity onset (Fig. 6b and Supplementary Movie 2). In addition, the threading two-point correlation function $\Phi(t, t_0)$ (defined in Methods) exhibits incomplete relaxation at all times, showing a dynamic threading steady state with persistent threadings (Fig. 6c).

Although in equilibrium the threading depth is correlated with the diffusion slow-down of individual rings[46], in the case of partly active rings, some specific shallow threadings (for example, see Fig. 7a) can significantly increase the relaxation time. The directionality of the rings tightens these threadings and they can be relaxed only if the active segment backtracks the passive tail. However, the directionality and the presence of other rings oppose the back-tracking. These threadings are likely members of the persistent class of threadings, but we could not determine the

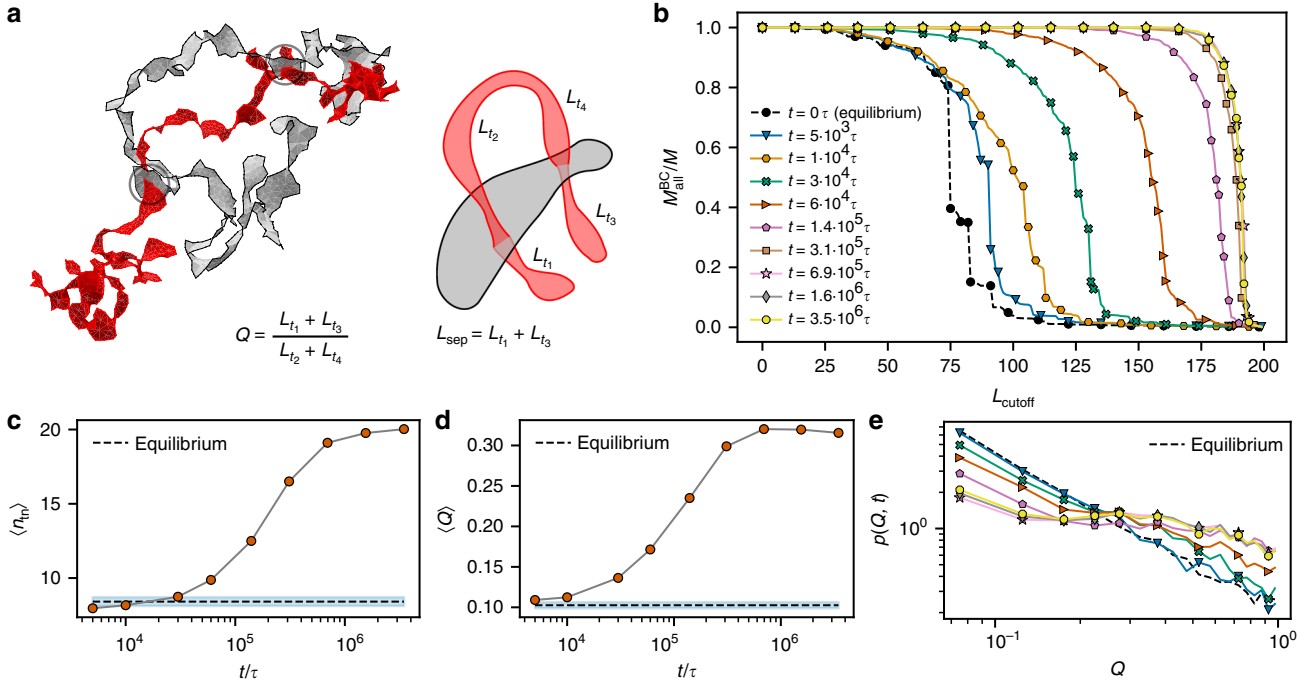

**Fig. 5 Threading analysis. a** The edgy image is an example of two rings in a steady state and their minimal surfaces (the circles mark the locations of threadings). The schematic smooth image shows the definition of $L_{sep}$ (measured in the number of monomers) and $Q$ (see text). **b** Relative size of the biggest threading cluster (the number of rings belonging to the biggest cluster, $M_{all}^{BC}$, divided by the total number of rings, $M$) as a function of the cutoff length $L_{cutoff}$. **c** Mean number of threaded neighbors $\langle n_{tn} \rangle$ by a ring as a function of time from the activity onset. The $n_{tn}$ is computed as the total number of threadings in the system $n_{th}$ divided by $M$. **d** Mean threading length ratio $Q$ as a function of time. **e** Distribution of $Q$ for different times after the activity onset (the legend is the same as in **b**).

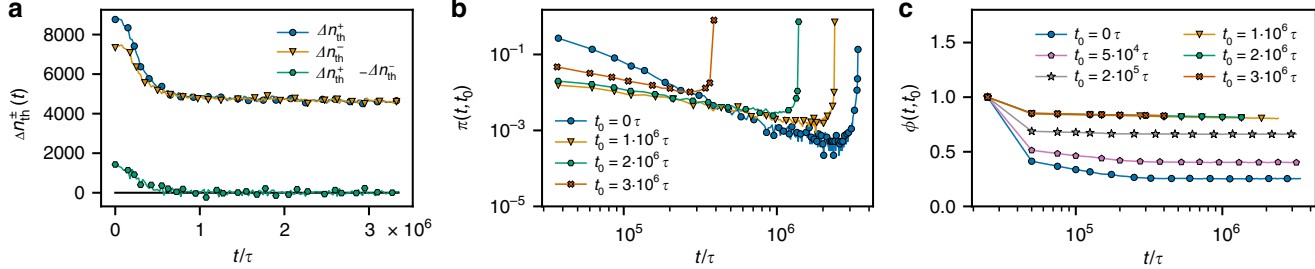

**Fig. 6 Threading dynamics. a** Threading gain $\Delta n_{th}^+$ and loss $\Delta n_{th}^-$ (see Methods for all definitions), as well as the net gain $\Delta n_{th}^+ - \Delta n_{th}^-$ as a function of time after the onset of activity. **b** Threading survival time distribution $\pi(t, t_0)$ measured from different $t_0$. The peak at late times represent the fraction of ring pairs threaded until the end of the analyzed data ($3.4 \times 10^6 \tau$). **c** Threading correlation $\Phi(t, t_0)$ as a function of $t$ measured from different $t_0$.

latter class yet. We suspect, that such threadings could also be relevant in the dramatic increase of the viscosity in stretched untangled melt of rings[48] (after this paper has been already accepted, the hypothesis has been confirmed in ref. [49]). To prove at least that ring topology and threadings are essential for the glass transition, we took a configuration of the system in the glassy state and cut the bond connecting two cold monomers in the middle of the cold segment on each ring. We further simulated the system which now consisted of $M$ linear triblocks of length $N$ with two cold segments at the ends and a hot segment in the center. The chain conformations change only moderately (Supplementary Note 5), but it is clear from $\langle g_3(t, t_0) \rangle / t$ (Fig. 7b) that the chains start to superdiffuse, releasing the accumulated mechanical stress (Supplementary Movie 3). Later on, the dynamics eventually crosses over to standard diffusion in analogy to orientational relaxation of self-propeled active particles[50]. As the non-topological properties of the system remained unchanged, we conclude that the phase segregation is not a

crucial element stabilizing the glass. The glassiness is driven by the enhanced threading due to the ring topology and the violated detailed balance[29].

## Discussion

We now return to the question of the order (continuous or type-A vs. discontinuous or type-B) of the glass transition for the system at hand. The usual control parameters driving vitrification in molecular or colloidal systems are the temperature and the density, and the typical glass-transition scenario there is discontinuous: the intermediate scattering function in the ergodic state develops a plateau, which grows in height and extends longer in time approaching the glass transition as the control parameters are changed[26,33,51]. In the continuous case, the non-ergodicity factor grows smoothly from zero to finite values. This second scenario is less common and its realization requires the presence of additional control parameters, such as porosity and randomness[36–38], tunable attraction widths in the

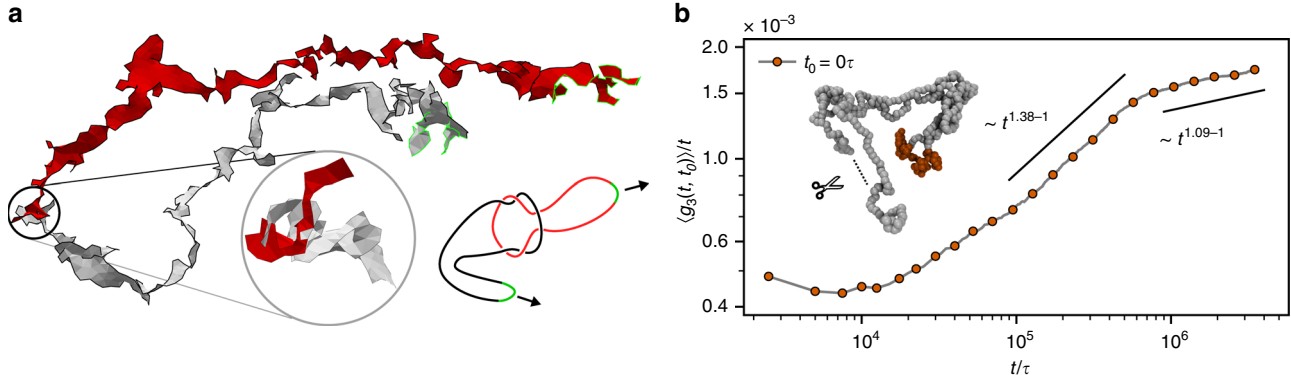

**Fig. 7 Tight threading and relaxation of cut rings. a** Two partly active rings with their minimal surfaces revealing a tight threading (detail in the inset). Active segments are marked green. Their sketched conformation shows how the directionality tightens the threading and introduces long relaxation. These as well as other threadings pose no long-time constraint if the rings were cut. **b** Mean squared displacement of the cut rings divided by time ($t$ is measured from the cutting event). The initial decrease at short times is the residual effect of the glassy state, until the former topological constraints get released and the centers of mass of the rings switch to superdiffusion at intermediate times. Later on, the chains cross over to the usual diffusion. Inset: snapshot of a cut ring at late times.

interactions[52,53] or confining periodic potentials[54] for which the amplitude and the wavelength can be independently varied. In such cases, lines of continuous glass transitions have been found in parameter space, and they are associated with higher-order singularities of the $A_3$- or of the $A_4$-type. The latter appear as endpoints of type-A transition lines that merge with type-B lines, as endpoints of type-B lines separating two glasses or as endpoints of $A_3$-lines in the latter case.

For the system at hand, several control parameters can be tuned: the fraction $M_a/M$ of partly active rings; the ratio $T_h/T_c$ of the temperatures of the hot and cold segments; the fraction $N_h/N$ of hot segments on a ring; and the number of monomers $N$ of the rings. The richness of the system makes the possibility of existence of higher-order singularities in principle possible. A detailed investigation in the vast space spanned by these is beyond the scope of this work; we focused mainly on the first two cases above. In Figs. 8a, b, we show the effect of gradually increasing the fraction of partly active chains, which induces a glassy state, as witnessed by the saturation of the mean-square displacement (Fig. 8a) and the growth of a non-ergodic plateau (Fig. 8b) as the ratio $M_a/M$ exceeds a number as small as 1/160. There is no evidence of the development of an intermediate plateau in the ergodic state preceding the glass, in full analogy with type-A transitions seen in the aforementioned systems[36–38,52–54]. The presence of a subdiffusive regime in the mean-square displacements (Fig. 8a) and of a logarithmic crossover intermediate scattering function (Fig. 8b) offer additional corroboration that the transition for this choice of the remaining system parameters is continuous, and thus higher-order singularities are present. We have found similar behavior (not shown) varying the ratio $T_h/T_c$. Naturally, this does not rule out that in other parts of the phase diagram the transition is governed by $A_2$-singularities and it is thus of B-type. This would give rise to a number of additional scenarios for the behavior of the relaxation functions, including the possibility of multiple relaxations observed in related models[54,55]. The presence of deep and tight threadings in our system bears an intuitive analogy with colloidal attractions[53], random pinning[36–38,51] or polymer-mediated bond formation[55,56] in systems featuring similar glass-transition phenomenologies. A detailed investigation of this issue, however, is a problem for the future.

The present active topological glass is remarkable by its distinct role of activity in comparison to the known active glasses composed of self-propelled particles[12,14]. There, the activity opposes

the glassiness as indicated by the increase of the effective temperature defined through a long-time limit of a generalized fluctuation–dissipation relation. At the same time, however, increasing the persistence time can lead to a decrease of the effective temperature and therefore promote the glass formation, and conversely, decrease in persistence time favors fluidization. In contrast, the microscopic model of the active topological glass has zero persistence time, but nevertheless drives the vitrification. This can be the consequence of the polymeric nature of the particles and the topological constraints that together create some persistence as illustrated by the superdiffusive regime. Moreover, the activity clearly causes increase of the number and the severity of topological constraints. This 'topological crowding' then can be viewed as a specific, effective attraction or pinning that further promotes the glass as mentioned above.

As regards the system with only a fraction of active rings present, the threading cluster analysis (more details in Methods and Supplementary Note 6) has revealed that it is always the active rings that are involved in the formation and maintenance of the largest cluster (Fig. 8c–e). The onset time of the glassy regime increases with decreasing $M_a$ (Fig. 8a). This delay is related to the slower building up of the largest cluster as fewer active chains participate (Supplementary Fig. 10). The ring length $N$ governs the number of topological interactions (see Supplementary Note 7 for varying ring length and Supplementary Fig. 12). Similar to linear polymer melts in equilibrium, due to extended chain configurations, each partly active ring overlaps with a number (extensive in $N$) of other rings. As a cooperative unthreading of rings is necessary for the relaxation to occur, we expect the relaxation time to grow at least exponentially with $N$[6]. Such a strong dependence is also known from the melts of polymeric stars, where it is due to the slow process of arm retraction that, however, can take place even without cooperative motion of the other chains[57].

While our model system is interesting from the fundamental physics point of view, it is also inspired from biology and may well bear important ramifications for the organization of the DNA fiber (chromatin) of higher eukaryotic cells. The large-scale static properties of the equilibrium melt of unknotted rings, such as the territorial organization, the scaling of $\langle R_g \rangle$ with $N$ or the so-called contact probability (Supplementary Fig. 1d) are consistent with the population-average conformation of the interphase chromatin[58]. This can be due to the common governing role of the topological constraints in both systems[58].

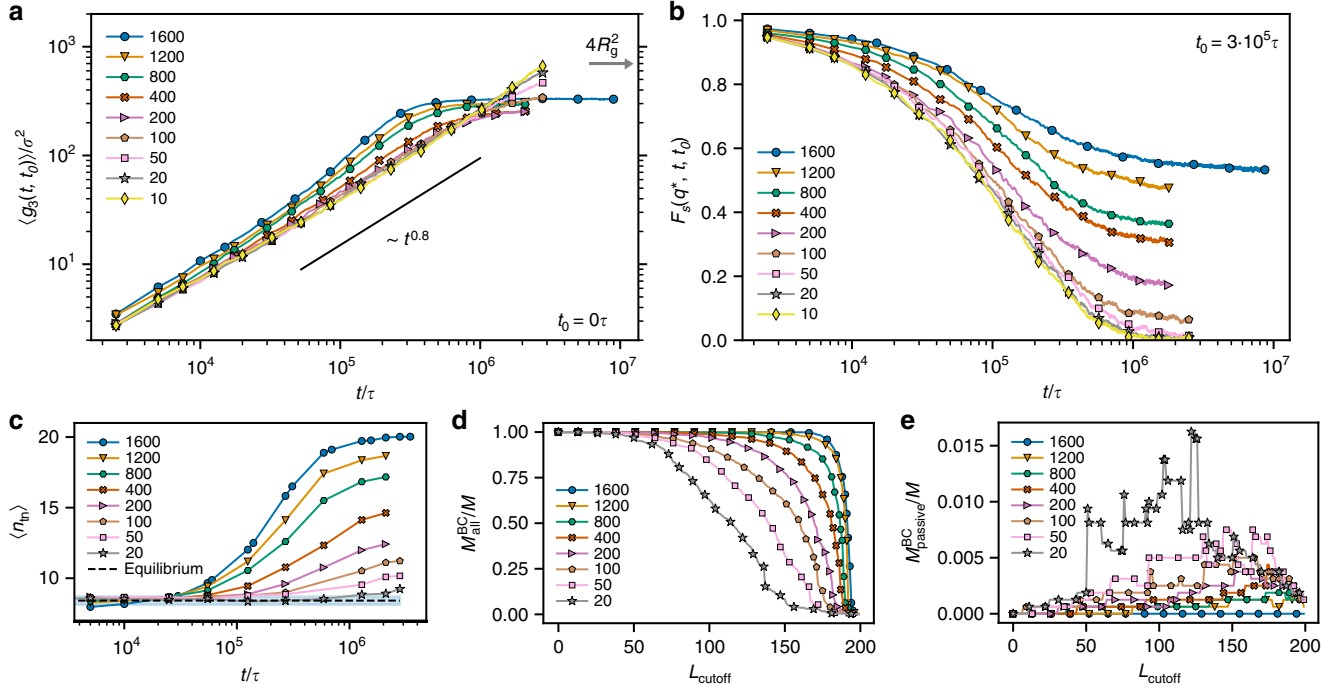

**Fig. 8 Varying the number of active rings. a** Mean squared displacement of the rings' centers of mass from the onset of the activity ($t_0 = 0$) as a function of time for different numbers $M_a$ of partly active rings indicated in the legend. **b** Self-intermediate scattering function $F_s(q^*, t_0)$ as a function of $t$ evaluated at $q^* \sigma = 0.35$ and $t_0 = 3 \times 10^5 \tau$ for different $M_a$. **c** Mean number of threaded neighbors $\langle n_{tn} \rangle$ by a ring as a function of time from the activity onset for different $M_a$ (see also Supplementary Fig. 8). **d, e** Threading cluster analysis for systems at with different number of partly active rings at $t = 2.1 \times 10^6 \tau$. **d** Relative size of the biggest cluster of those clusters that contain any kind (active and passive) of rings. **e** Relative size of the biggest cluster of those clusters that contain only passive rings (see Methods for details). Note the difference in the scale of the ordinate for the passive ring clusters.

The rationale behind the ring model is the timescale separation— the chromosomes are linear chains and as such equilibrate and tangle by reptation, however, for the length and density of chromatin, such relaxation would take significantly longer than the cell's life time[59]. Therefore, the constraints arising from the uncrossability of the chains can be modeled as permanent on biological time scales. This is done effectively by the closure of the chain's ends which inhibits the reptation. Furthermore, the chromatin association with the nuclear lamina[60] hinders reptation relaxation and the chromatin can be viewed as loops between lamina contact points. In contrast, the topoisomerase II enzyme can resolve the topological constraints, by cutting the fiber, passing segment through sealing up the cut back[61], although the extent to which it affects global topology on biological time scales in interphase is unclear. Alternatively, the rings can also represent small scale chromatin loops extruded or maintained by Structural Maintenance of Chromosomes protein complexes[62,63], and/or Topologically Associating Domains[64] that do not link. Current experimental evidence for the topological state (knottedness) of the chromatin fiber also varies. While conformations inferred from contact probability measurements exhibit knots[65], the knot analysis[66] of the direct observations of fluorescently labeled chromatin segments finds mostly unknotted segments. Unable to refine the scales of the topological constraints completely at this point, we assumed their existence for the typical observation times and examined the consequences. Therefore, we mention phenomena on chromatin at various scales, for which the interplay of the topology and the activity could be relevant.

The segmental activity with thermal-like fluctuation spectra has been measured to be stronger in the normal living cell nuclei[67] as opposed to energy-deprived cells, and has its origins in the energy dependent processes, such as chromatin repair or remodeling, DNA transcription, or loop extrusion. We conjecture

that the phenomena observed in our partly active system could also be relevant in biological context, on the basis of the following similarities with our model: genes exhibit size increase upon transcription decoupled from the chromatin decondensation[68]; a highly transcribed gene shows directed motion[69]; overall chromatin loci exhibit heterogenous subdiffusive dynamics[67] reminiscent of glassy behavior[23,70]; the active and inactive chromatin are spatially segregated[71]; and the chromatin exhibits a doubly folded structure at small scales[72]. Naturally, these effects have also alternative explanations. The phase separation and glassy dynamics could be observed for a copolymer models where different chromatin segments have different interaction potentials based on their epigenetic state[23,70,73,74] or by interaction with binding proteins[74,75]. The double folded structure is likely to be attributed to supercoiling due to the torsional stress induced on the fiber in the process of transcription or loop extrusion[62,76,77]. Nevertheless, we show that the activity in combination with topological constraints at the fiber level can complement the observed phenomena and should be considered in a more complete picture of the chromatin organisation[78,79]. As the models above typically do not consider topological constraints and activity, our findings represent a completely new mechanism for the observed phenomena. Certainly, more work is required to determine the relative contributions of the different mechanisms in the various cases. Potentially, the different nature of the observed glassy states, namely the monomer glass due to caging in copolymer models[23,70] and topological glass here, could be used to discern which one could be present or dominant in the chromatin context.

Activity complemented with topological constraints at the microscopic level can lead to rich system dynamics. We have demonstrated this on a model system of dense, unknotted and nonconcatenated ring polymers with active segments. First, the

directionality of polymers in a dense environment arises from the topological constraints and the isotropic noises of unequal strength. Similar to ref. [2], the superdiffusive motion is connected to a major particle deformation at high density. In contrast, however, in our work, the superdiffusivity is triggered by the active noise. Second, a novel state of matter—the activity-driven topological glass—arises from the activity-enhanced ring threading. In contrast to the conjectured equilibrium topological glass[5], our present model allows the creation of a topological glass for rings of accessible lengths ($7N_e$) using activity. Moreover, only a low number of partly active rings is neccesary, making the present model suitable for experimental testing with extracted bacterial DNA[80] or synthetic ring polymers[3], driven by molecular motors fueled by ATP hydrolysis. The effective temperature ratio $T_h/T_c$ of a factor of three that we used here is within reach, since ATP hydrolysis releases more than $10k_BT$[81]. However, as our preliminary results suggest, even smaller temperature ratios can be sufficient. Other means of selective heating could be attempted by fluctuating external fields or infrared irradiation selectively coupling to individual groups in the polymer. The latter mechanism can yield a fluid material with reversible vitrification upon light exposure. What is the proper topological order parameter of the glass transition, or what are the fragility properties of the active topological glass, are just a few intriguing questions to be addressed in the future. The present work paves the way for a development and investigation of these novel, molecular, topology-based, responsive materials.

## Methods

**Model.** All particles interact via a purely repulsive Lennard-Jones potential

$$U_{LJ}(r) = \left(4\varepsilon\left[\left(\frac{\sigma}{r}\right)^{12} - \left(\frac{\sigma}{r}\right)^6\right] + \varepsilon\right)\theta(2^{1/6}\sigma - r), \quad (1)$$

where $\theta(x)$ is the Heaviside step function. The connectivity of polymers is maintained by the finitely extensible nonlinear elastic potential

$$U_{FENE}(r) = -\frac{1}{2}r_{max}^2 K \log\left[1 - \left(\frac{r}{r_{max}}\right)^2\right], \quad (2)$$

where $K = 30\varepsilon/\sigma^2$ and $r_m = 1.5\sigma$. These parameters make the chains essentially uncrossable. The angular potential is

$$U_{angle} = k_\theta(1 - \cos(\theta - \pi)) \quad (3)$$

with $k_\theta = 1.5\varepsilon$.

The studied systems are monodisperse at fixed volume with the total monomer density $\rho = 0.85\sigma^{-3}$. We used very large systems of $M = 1600$ chains to avoid unphysical self-threadings of extended rings due to periodic boundary conditions. As shown in Supplementary Fig. 1c, our systems are large enough to assure that the rings are smaller than the simulation box at all times. All the simulations were performed at constant volume with two Langevin thermostats using the large-scale atomic/molecular massively parallel simulator (LAMMPS) engine[82]. The equations of motion were integrated with the time step $\Delta t = 0.005\tau$, where $\tau = \sigma(m/\varepsilon)^{1/2}$.

To model heterogeneous activity of the rings, on each polymer a consecutive segment of length $N_h = N/8$ monomers is considered active by subjecting it to stronger thermal-like fluctuations (isotropic, uncorrelated, white noise) in comparison to the rest $(N - N_h)$ monomers of the ring that remain passive. In particular, the active monomers are connected to a Langevin thermostat of temperature $T_h = 3$ (units of $\varepsilon$ and the Boltzmann constant set to unity are being used throughout), while passive ones are coupled to a second Langevin thermostat with temperature $T_c = 1$. The coupling constants of both thermostats are $\gamma = (2/3)\tau^{-1}$. From our earlier study[83] we know that such values of $\gamma$ and $T_h$ lie in the range of the onset of a non-equilibrium microphase separation in the active-passive mixtures of polymers of 40 monomers, but colloidal systems would not demix[29,31]. Our preliminary studies show that even a weaker temperature contrast is sufficient for the glassy state to occur. We leave a more detailed characterization of the phase diagram for the future study.

**Model details.** The employed model was frequently used not only for the melt of linear chains but also for the melt of rings[27,28] at $T = 1$. Therefore, we already know a range of useful properties of the equilibrium system, such as the entanglement length $N_e = 28 \pm 1$ and the typical diffusion times of the rings. In equilibrium, the diffusion time $\tau_{diff}(N)$ of a ring is defined as the mean time required

for its center of mass to diffuse over $2R_g$, where

$$R_g \equiv \langle R_g^2\rangle^{1/2} = \left\langle\frac{1}{N}\sum_{i=1}^{N}(\mathbf{r}_i - \mathbf{R})^2\right\rangle^{1/2} \quad (4)$$

is the mean radius of gyration. Above, $\mathbf{r}_i$ denotes the position of the $i$-th monomer and $\mathbf{R}$ is the position of the center of mass of the ring. The angles $\langle\cdots\rangle$ stand for the average over different chains. As reported in refs. [27,28], $\tau_{diff}(N = 400) \simeq 4 \times 10^5\tau$.

To quantify dynamical evolution of the system, we consider squared displacements of the centers of mass of the rings, $g_3(t, t_0)$, at a given time $t$ provided that the measurement started at $t_0$ ($t_0 = 0$ is the activity onset, or, in the case of cut rings, the moment of the cutting):

$$g_3(t, t_0) = [\mathbf{R}(t_0 + t) - \mathbf{R}(t_0)]^2, \quad (5)$$

where $\mathbf{R}(t)$ is the position of the center of mass of a ring at time $t$ with respect to the center of mass of the whole system at that time. Typically, we report the mean squared displacement $\langle g_3(t, t_0)\rangle$ averaged over the rings only, i.e., without additional averaging over multiple time origins as the dynamics of the system are not stationary in general.

The relaxation dynamics of the system is also characterized by the self-part of the intermediate scattering function defined as

$$F_s(q, t, t_0) = \frac{1}{M}\sum_{m=1}^{M}\exp(i\mathbf{q}\cdot(\mathbf{R}_m(t_0 + t) - \mathbf{R}_m(t_0))), \quad (6)$$

where $\mathbf{R}_m$ is the position of the center of mass of the $m$-th ring. We evaluated it at $q^*\sigma = 0.35$ corresponding to the maximum of the static structure factor (see Supplementary Fig. 6 for other $q$-values).

We define ergodicity breaking parameter as

$$EB(t) = \frac{\langle g_3(t, t_0, \Delta)^2\rangle - \langle g_3(t, t_0, \Delta)\rangle^2}{\langle g_3(t, t_0, \Delta)\rangle^2}. \quad (7)$$

Here, $g_3(t, t_0, \Delta)$, being mainly a function of the lag time $\Delta$, represents the time average:

$$g_3(t, t_0, \Delta) = \frac{1}{t - \Delta}\int_{t_0}^{t_0 + t - \Delta}[\mathbf{R}(t' + \Delta) - \mathbf{R}(t')]^2 dt' \quad (8)$$

where $\mathbf{R}$ is the position of the ring's center of mass with respect to the global center of mass.

**Threading analysis.** The minimal surfaces are spanned on the fixed contours of the rings from the molecular dynamics simulations and then minimized using the overdamped surface tension evolution under the constraint of fixed disc topology. We used the Surface Evolver software[84] and followed the protocol in[47] ensuring that for these ring lengths the area of the final surface is close to the minimum.

The separation length is defined as

$$L_{sep} = \min\left(\sum_{i=even}L_{t_i}, \sum_{i=odd}L_{t_i}\right), \quad (9)$$

where $L_{t_i}$ is the (threading) length between the $i$-th and the $(i + 1)$-th penetrations of the surface (see Fig. 5a). $L_{sep}$ approximates how much of the threading rings material is on one side of the threaded ring. The approximation lies in the assumption that two following surface piercings are in the opposite direction with respect to the surface normal vector. In many cases, there are only two surface penetrations between two rings (see Supplementary Fig. 3) and in such case the approximation is exact because of the non-concatenation constraint of the rings conformations.

The threading cluster analysis is performed in the following way. At first, each ring is analyzed for threadings with other rings. Two rings are assigned to the same cluster if at least one of them threads the other one with $L_{sep} > L_{cutoff}$. Then, the size (in terms of the number of members) of each cluster is determined and the biggest cluster is found. For mixtures of active and passive rings the size of the biggest cluster containing any kind of rings is considered (for example, Fig. 8b) and separately the biggest cluster of those clusters that contain only passive rings is calculated, such as in Fig. 8c.

To characterize the dynamics of threadings we compute the threading gain $\Delta n_{th}^+(t)$ and loss $\Delta n_{th}^-(t)$ as function of time. To do so, we compare the set $T(t)$ of pairs of rings that are threaded at time $t$ with the set $T(t + \Delta t)$ of threaded pairs at time $t + \Delta t$. Then $\Delta n_{tn}^+(t) = |T(t + \Delta t) - T(t)|$ and $\Delta n_{tn}^-(t) = |T(t) - T(t + \Delta t)|$, where $|x|$ is the number of members of set $x$. Note that $\Delta n_{th}^\pm(t)$ is different from the derivative of the number of threaded neighbors, because there exist a state with the derivative zero, but high threading exchange, which is exactly the case in the steady state. For all calculations of the threading dynamics, we used $\Delta t = 2.5 \times 10^4\tau$ and maximum simulation time $3.4 \times 10^6\tau$.

A related quantity characterizing the threading correlation is $\Phi(t, t_0) = \frac{1}{n_{tn}(t_0)}\sum_{i,j=1}^{n_{tn}(t_0)}T_{ij}(t_0)T_{ij}(t_0 + t)$, where $T_{ij}(t)$ is one, if threading between rings $i$ and $j$ exists at time $t$ and zero otherwise, and $n_{tn}(t)$ is the number of

threaded neighbors at time $t$. The $\Phi(t, t_0)$ gives the time correlation function for the existence of threadings.

In addition, we compute the threading survival time distribution $\pi(t, t_0)$ measured from different $t_0$. To construct $\pi(t, t_0)$, we look at all threaded ring pairs at time $t_0$ and track how long they remain threaded before they unthread for the first time.

## Data availability

The relevant data sets generated during and/or analyzed during the current study are available from the corresponding authors on reasonable request.

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

## Acknowledgements

J.S. acknowledges support from the Austrian Science Fund (FWF) through the Lise-Meitner Fellowship No. M 2470-N28. The authors would like to acknowledge networking support by the COST Action CA17139. We are grateful for a generous computational time at Vienna Scientific Cluster and Max Planck Computing and Data Facility. This work has been supported by the European Research Council under the European Union's SeventhFramework Programme (FP7/2007–2013)/ERC GrantAgreement No. 340906-MOLPROCOMP.

## Author contributions

J.S. and K.K. designed the research with the contributions from I.C. and C.N.L. J.S. and I.C. performed the simulations and data analysis. J.S., I.C. and C.N.L. interpreted the results. J.S. wrote the paper with the contributions of I.C., C.N.L. and K.K.

## Competing interests

The authors declare no competing interests.
