## [Peer Review File · Nature Communications]

Reviewers' comments:

Reviewer #1 (Remarks to the Author):

The article titled "Active topological glass" by Smrek et al presents an intriguing piece of work and I think this paper should eventually be accepted in Nature Communications. However, I have a number of comments that the authors should address before I can recommend it for publication:

(1) Writing introduction of a paper on active glass is complicated since one first needs to address the works on equilibrium glasses and then the works on active glasses of various types. Given this complexity, the authors seem to have done a commendable job. However, there are some comments:

(i) Paragraph 3, Page 1: While addressing the works on self-propelled particles, they write " ... can even promote the glass formation". This is a bit misleading in the sense that activity in these systems never promotes glassiness compared to the passive system. Consider the passive system in glassy regime, as soon as one introduces activity of the sort of self-propulsion, the system becomes "less-glassy" compared to the passive system irrespective of the self-propulsion parameters. In these systems, activity promotes glassiness in the following sense: if one takes self-propulsion in the form of Ornstein-Uhlenbeck active noise, the system goes towards the glassy state if one increases the persistence time compared to when persistence time was smaller. This should be clarified in the text.

(ii) The last sentence in this paragraph also requires rephrasing. Note that the active glass differs from the thermal equilibrium glass. If one calculates the effective temperature as the ratio of response and correlation functions, this quantity is time-dependent in the active system whereas it is constant (and equals the temperature) in equilibrium system. The reference [Nandi and Gov, Soft Matter, 2017,13, 7609-7616] should also be cited in this context.

(iii) In connection to my first comment, the authors are showing results which are quite different from what is known from glassy systems of self-propelled particles. Their activity has zero persistence time (in the language of self-propulsion), however, the activity promotes glassiness compared to the passive system. Of course, their system is different, however, the role of activity is quite different and I think this difference should be pointed out at this point of the paper.

(iv) The authors are discussing some topological glasses and the works of Bi et al, where they looked at the glassy dynamics in vertex models, should be mentioned. Here are some of the relevant references: [Bi D, Lopez JH, Schwarz JM, Manning ML (2015) A density-independent rigidity transition in biological tissues. Nat. Phys. 11:1074], [Bi D, Yang X, Marchetti MC, Manning ML (2016) Motility-driven glass and jamming transitions in biological tissues. Phys. Rev. X 6:021011] etc.

(2) In the results section, Page 3: The argument of why the hot segment is dragging the cold segment should be clearer. I think the message is fine, but the wording could be better. Also some typo: "The tail than ... "  "The tail then ...".

(3) In page 4: I have several comments on this subsection on glassy behaviour. I think the dynamics is well-characterized to point out their main result that it is indeed glassy. However, there are more in the results that the authors should analyze:

(i) The two-time correlation functions do not show the characteristic two-step relaxation scenario of ordinary glasses. The authors also comment "... we can not observe the second relaxation stage". However, such relaxation scenario, as reported in Fig. 3(a) is not unusual in glassy systems, although

they require more conditions. In the language of mode-coupling theory of glasses, they are known as the A3 (or continuous) critical point (the ordinary bulk glass is A2 (or discontinuous) critical point). In physical scenarios like confinement [Phys. Rev. E 84, 061501 (2011)], pinning [Phys. Rev. E 75, 031503, 2007] or under external potential, even simple atomic glassy systems can show relaxation scenarios like this. The complexity of the system the authors are considering means that it is quite possible that the system is close to such a continuous critical point [see for a definition: Phys. Rev. Lett. 113, 245701 (2014)].

My guess from looking at the results is that the system is close to the A3 critical point and the second relaxation will never be observed no matter how long they run the simulation. However, this is only my guess and need not be true. It is important that the authors include a discussion on this point, otherwise the obvious question that arises is why there is no two-step relaxation of the correlation function and this second relaxation (alpha relaxation) must be shown, even if for at least one measurement.

(ii) Their results clearly show there is aging in the system. This is not surprising since the protocol the authors are following should naturally give rise to aging. The quench in this system is in activity and not temperature or density [Phys. Rev. Lett. 109, 115702 (2012)]. This is interesting from the glass perspective as well. In my opinion, the authors should refrain from making comments like "... t_0 plays the role of the volume fraction ...". One can probably view the system in terms of an evolving effective volume fraction, but I think this is oversimplifying the main perspective of aging.

(iii) In an aging system, key characteristics of the system changes with the aging time t_0 . It will be important to show a more quantitative characterization of this aspect. The authors could show how relaxation time and the non-ergodicity parameter, $f(q, t_0)$, change with t_0 . Then consider two different quenches, for example for two different number of active rings, and show if these functional dependences remain same. Do the exponents change from those of a passive system?

(4) Fig. S2(d): Could the authors show it in logarithmic scale? I would expect a similar behavior as $F_s(q, t, t_0)$. The reason is related to some of the theoretical approaches and not particular to the current system; in the glassy regime, relaxation of any two-point quantities should show similar behavior. Of course this is an approximation, but it will be nice to see if that holds in this system as well.

Reviewer #2 (Remarks to the Author):

In this work Smrek and colleagues build upon their recent work on dense solutions of linear polymers with different temperatures and this time consider ring polymers. This seemingly small change has a big impact on the dynamics of the system and indeed they provide computational evidence for the onset of an "active topological glass" triggered by activity. They extensively characterise this state and, finally, they draw a connection with the behaviour of chromatin in vivo.

I think that this is a very fundamental piece of work in several respects and I strongly support its publication in Nature Communication. I believe it will be considered seminal in both polymer physics and active matter (and perhaps even in genome organisation) communities.

At the same time, I also feel that some sections may be improved or made more precise as follows:

1. "The rings cannot cross each other, but are known to thread — one ring piercing through an eye of another ring (see Fig. 5a), which temporarily topologically constraints the motion of the two rings". I

would argue that it is only the passively threaded ring whose motion is constrained.

2. "... which would give rise to exponentially long relaxation times".

If there is a critical parameter for which vitrification occurs, then the rise of relaxation time is expected to be super-exponential (or exponential with divergence, see VFT law for glass-forming systems) which is seen in the case of pinning.

Otherwise, if it is a cascade of relaxations then one may expect just a steep power-law (?).

With this comment, I just want to point out that it is not clear (at least to me) that the relaxation due to hierarchical threadings necessarily leads to an exponentially long relaxation. Can the authors comment?

Maybe it would be interesting to quantify the relaxation time of the rings in their simulations by measuring the effective diffusion coefficient against the ratio of hot/cold temperatures. I understand that this would require a whole set of new simulations and since there are already enough results in this paper perhaps the authors may want to consider this suggestion for the future.

3. Just to iterate on the point above.

The authors mention arm retraction in page 7 and argue that the relaxation of stars and unthreading are similar in nature. I disagree with this in the sense that different arms of a star are independent, but any section of a ring contour is directly connected to any other. Rings lack branching points which decouple the relaxation of different segments. So even if a section of a ring is threading a neighbour, then the diffusive motion of the *non-threading* section affects the other and speeds up the unthreading by "dragging" it away. Perhaps the authors can comment on this?

4. "Then, in the limit of long rings and high density a glassy behavior can be extrapolated to $f \rightarrow 0$ ".

Actually it is the other way around: the extrapolation (done using any range of chain length and density) to $f \rightarrow 0$ is done to find the critical values (of length and density) for which pinning is no longer required.

5. "...it is unlikely to be utilized experimentally to drive the topological glass transition". This is a strong statement which is unnecessary and unsupported. For sure, the proposed activity creates a completely different (and perhaps better) route to achieve topological freezing but one can still envisage a number of (experimentally challenging) set-ups in which a fraction of the rings is slower than the other and thus impose an effective pinning...

6. The authors state that the temperature for the hot regions is 3 times larger than that for the cold ones. This can hardly be considered "modest" to my opinion. What is the rationale for choosing this number (apart from driving the microphase separation which in any case is unrelated to the glassy dynamics)?

It also begs the question of how can one realise such a strong difference experimentally...

I think it would be useful if the authors could perhaps add a sentence or two on this (experimental realisations) in the conclusions.

7. By looking at fig.5c I get the impression that, effectively, turning on the activity creates a bias towards creating threadings rather than removing them; in other words, the rates of threading/unthreading are no longer equal (as they are expected to be in equilibrium for detailed balance) and are skewed by the activity.

My question to the authors is: do you think the long-time value you observe is because the system at that point is glassy and cannot create new threadings anymore or is it a new steady state? i.e., do threadings still exchange?

7a. Iterating on 7, do you think that if it was possible to simulate (or realise) rings that display about

20 threadings per ring *in equilibrium*, one should expect glassy dynamics without either pinning nor activity?

In other words, is 20 threadings another "magic number"?

I am saying this because 20 is also the "magic" Kavassalis–Noolandi number to which the overlap parameter converges to for large rings (see Rosa&Everaers PRL 2014 and Ge et al Macromol 2016). I wonder if this is only a coincidence or if one needs, in practice, to thread all neighbours in order to vitrify.

7b. Iterating on 7 and 7a, can the authors measure and report the average number of threadings per chain in the systems with varying number of active chains (Fig. 7)?

Do the systems need to reach $\langle n_{\{tn\}} \rangle \geq 20$ in order to freeze?

On the biology side:

8. I have never been a fan of the mapping ring polymers -> chromatin in vivo. I think it effectively works well, but we still do not understand why. In particular there is one strong argument against it: the fact that the topological constraints in the two systems are completely different.

In vivo, our chromosomes are not only not rings (some relatively small sections of them may topologically look like rings, e.g. looped TADs) but are also constantly cut and glued back by Topoisomerases. The latter entails that there is no global topological invariant that is preserved, oppositely to standard melts of rings which are simulated here and elsewhere.

9. I acknowledge that the effectively "hot" regions in the authors' simulations are good representations of transcribed genes (and other active sections of the genome) *but* the segregation of active/inactive compartments and heterogeneous and subdiffusive (or glassy) dynamics can also be explained by (and perhaps more simply) co-polymer models (Jost et al NAR 2014, Michieletto et al PRX 2016), bridge/binding proteins (Shi et al, Nat Comm 2018) or even just confinement (Kang et al PRL 2015).

It is unclear, from the way this section is written, if the authors imply that modellers should model the genome as a ring or if they should include regions with differential activity (or both).

10. Finally, in this paper the authors find glassy behaviour due to threadings, but these constraints are easily resolved by TopoII in vivo. So perhaps the origin of slow dynamics of chromatin must be found elsewhere rather than threadings (see point 9).

With these comments I don't intend to put off the authors from drawing this connection which is very fascinating, but perhaps they should be more careful with their wording and try to give a more complete and accurate picture (albeit this is far from complete yet and so not an easy task!).

REPLY TO REVIEWER 1

The article titled "Active topological glass" by Smrek et al presents an intriguing piece of work and I think this paper should eventually be accepted in Nature Communications. However, I have a number of comments that the authors should address before I can recommend it for publication:

Reply: We thank the reviewer that s/he has recognized our work as intriguing and worthy publication.

(1) *Writing introduction of a paper on active glass is complicated since one first needs to address the works on equilibrium glasses and then the works on active glasses of various types. Given this complexity, the authors seem to have done a commendable job. However, there are some comments:*

(i) *Paragraph 3, Page 1: While addressing the works on self-propelled particles, they write " ... can even promote the glass formation". This is a bit misleading in the sense that activity in these systems never promotes glassiness compared to the passive system. Consider the passive system in glassy regime, as soon as one introduces activity of the sort of self-propulsion, the system becomes "less-glassy" compared to the passive system irrespective of the self-propulsion parameters. In these systems, activity promotes glassiness in the following sense: if one takes self-propulsion in the form of Ornstein-Uhlenbeck active noise, the system goes towards the glassy state if one increases the persistence time compared to when persistence time was smaller. This should be clarified in the text.*

Reply: Indeed the glassiness promoted by the longer persistence time is what we had in mind in that paragraph. We agree that this should be spelled out explicitly and we have done so in the updated manuscript:

While, intuitively, activity opposes glassiness by enhancing mobility of the particles, some active models can exhibit a more complex behavior as function of the active control parameters. For example, increasing the persistence time of the active Ornstein-Uhlenbeck particles can either glassify or fluidize the active system, depending on the particular state point, as a result of nontrivial velocity correlations in the system [12-15]

(ii) *The last sentence in this paragraph also requires rephrasing. Note that the active glass differs from the thermal equilibrium glass. If one calculates the effective temperature as the ratio of response and correlation functions, this quantity is time-dependent in the active system whereas it is constant (and equals the temperature) in equilibrium system. The reference [Nandi and Gov, Soft Matter, 2017,13, 7609-7616] should also be cited in this context.*

Reply: We thank the reviewer for pointing this out and we agree we should be more specific. We have rephrased the sentence and added more details in order to briefly state the differences and similarities of the active and passive glasses.

Indeed, some system properties, such as the time-dependent effective temperature are pertinent to active fluids and render also the corresponding glass transition distinct from the passive one. In particular, the location and the existence of the glass transition of active fluids are dependent on the microscopic details of the activity mechanism. Nevertheless, close to the transition region, universal features of the passive glassy dynamics have been found recently for active spin-glasses [11] and self-propelled particles in the nonequilibrium mode-coupling theory (MCT) [15]. For instance, the scaling of the relaxation time with activity control parameters is governed by the same exponent as in the passive MCT.

(iii) *In connection to my first comment, the authors are showing results which are quite different from what is known from glassy systems of self-propelled particles. Their activity has zero persistence time (in the language of self-propulsion), however, the activity promotes glassiness compared to the passive system. Of course, their system is different, however, the role of activity is quite different and I think this difference should be pointed out at this point of the paper.*

Reply: This is a very good point and we thank the reviewer for bringing it up. We highlight and discuss this fact now in the the updated manuscript. Although the microscopic activity model has no persistence, the polymeric nature of the particles and the topological constraints generate, at least temporarily, a non-zero persistence as illustrated by the superdiffusive regime of the mean square displacement. We think that this effect can be related to the promotion of the glassiness. Additionally, our system has no attractive interaction, but the activity clearly causes increase in the number and the severity of the topological constraints. This "topological crowding" then can be viewed as a specific, effective attraction or pinning further promoting the glass. This is also in line to our findings related to question 3*i* and has been accounted for in the updated manuscript as follows:

The present active topological glass is remarkable by its distinct role of activity in comparison to the known active glasses composed of self-propelled particles [13,15]. There, the activity opposes the glassiness as indicated by the increase of the effective temperature defined through a long time limit of a generalized fluctuation-dissipation relation. At the same time however, increasing the persistence time can lead to a decrease of the effective temperature and therefore promote the glass formation, and, conversely, decrease in persistence time favours fluidization. In contrast, the microscopic model of the active topological glass has zero persistence time, but nevertheless drives the vitrification. This can be the consequence of the polymeric nature of the particles and the topological constraints that together create some persistence as illustrated by the superdiffusive regime. Moreover, the activity clearly causes increase of the number and the severity of topological constraints. This “topological crowding” then can be viewed as a specific, effective attraction or pinning that further promotes the glass as mentioned above.

- (iv) *The authors are discussing some topological glasses and the works of Bi et al, where they looked at the glassy dynamics in vertex models, should be mentioned. Here are some of the relevant references: [Bi D, Lopez JH, Schwarz JM, Manning ML (2015) A density-independent rigidity transition in biological tissues. Nat. Phys. 11:1074], [Bi D, Yang X, Marchetti MC, Manning ML (2016) Motility-driven glass and jamming transitions in biological tissues. Phys. Rev. X 6:021011] etc.*

Reply: We agree that these are relevant works for the manuscript in the relation of the glassy dynamics to the biological context of our work. Although there is no direct evidence of relevance of topology in these papers, the rigidity transition described occurs at constant high density similarly to our case. In the updated manuscript we add a paragraph on the biological relevance of glass, jamming and rigidity transitions in active systems and cite these works.

Glassy and heterogeneous dynamics appears in a range of biological situations ranging from the bacterial cytoplasm [22] to collective cell migration in tissues [23]. While on the subcellular level, glass-like properties have been attributed to the high crowding, size and interaction heterogeneity of the constituents [22,24], the confluent tissues modeled by vertex models exhibit a new type of rigidity transition at constant density without [25] and with active motion [26], attributed to an interplay of cells’ shape and persistence of motion. Although there is no topology involved in these models and, therefore, they are inherently different from our present work, the transition occurs due to shape changes at constant density similarly to our case. After detailed account of the physics of the active topological glass, we discuss its relevance for the organization and dynamics of chromosomes in living eukaryotic cells.

- (2) *In the results section, Page 3: The argument of why the hot segment is dragging the cold segment should be clearer. I think the message is fine, but the wording could be better. Also some typo: "The tail than ... " → "The tail then ...".*

Reply: We have acknowledged this point and clarified the discussion about the mechanism of the observed transient superdiffusion on p. 4 of the revised manuscript:

The superdiffusive regime is a consequence of the specific non-equilibrium dynamics through a mesh of topological constraints. As the detailed balance is violated due to the coupling to different thermostats, the pulling of the active segment forwards is stronger than the pulling of the cold tail backwards. The hot segment robustly explores new sites that are spontaneously freed due to density fluctuations and progressively drags behind itself the cold tail (see Supplementary Video 1). Furthermore, such motion of the active segment away from its cold tail through the environment of neighboring rings introduces new topological constraints that the cold tail must obey. These constraints restrict the transversal motion of the chain. Finally, when the chain is getting more stretched some time after the activity onset, the motion of the hot segment backwards to the cold tail is compromised by the chain flexibility and, therefore, the motion away from the tail prevails. As a result, the cold tail follows the hot head slowly, but ballistically along a trail imposed by the topologically constrained neighboring rings. At these length scales, the trail is characterized by the size R of the static conformation of the tail and it scales with the contour distance s as $R(s) \sim s^{\nu_{\text{trail}}}$ where the exponent $\nu_{\text{trail}} = 0.57$ (Fig. 2d). Therefore, the directed dynamics ($s \sim t$) along such contour is superdiffusive with $\langle g_3(t, t_0) \rangle \sim R^2(s(t)) \sim t^{2\nu_{\text{trail}}}$, which is in agreement with our observation (Fig. 2c). Moreover, the onset time of the superdiffusion $t \approx 10^5 \tau$ is consistent with the onset time of a configuration that is more open than a random walk, that is $\nu_{\text{trail}} > 0.5$ as seen in Fig. 2d, and the dragging mechanism is consistent with the fact that the end points of the active segment are closer to each other than in the equilibrium case (Fig. 2e).

- (3) *In page 4: I have several comments on this subsection on glassy behaviour. I think the dynamics is well-characterized to point out their main result that it is indeed glassy. However, there are more in the results that the authors should analyze:*
- (i) *The two-time correlation functions do not show the characteristic two-step relaxation scenario of ordinary glasses. The authors also comment "... we can not observe the second relaxation stage". However, such relaxation scenario, as reported in Fig. 3(a) is not unusual in glassy systems, although they require more conditions. In the language of mode-coupling theory of glasses, they are known as the A3 (or continuous) critical point (the ordinary bulk glass is A2 (or discontinuous) critical point). In physical scenarios like confinement [Phys. Rev. E 84, 061501 (2011)], pinning [Phys. Rev. E 75, 031503, 2007] or under external potential, even simple atomic glassy systems can show relaxation scenarios like this. The complexity of the system the authors are considering means that it is quite possible that the system is close to such a continuous critical point [see for a definition: Phys. Rev. Lett. 113, 245701 (2014)].*

My guess from looking at the results is that the system is close to the A3 critical point and the second relaxation will never be observed no matter how long they run the simulation. However, this is only my guess and need not be true. It is important that the authors include a discussion on this point, otherwise the obvious question that arises is why there is no two-step relaxation of the correlation function and this second relaxation (alpha relaxation) must be shown, even if for at least one measurement.

Reply: We thank the reviewer for raising this point and for bringing to our attention the possibility of a continuous glass transition, associated with higher-order singularities. Motivated by this, we have performed additional simulations varying the fraction of partly-active chains as well as the ratio T_h/T_c between the temperatures of the hot and the cold segments, to approach the glass transition point(s) more accurately and obtain information on the nature of the glass transition. There is indeed evidence suggesting that there are parts of the phase diagram in which the transition is continuous, though our study is not and cannot be comprehensive or exhaustive at this stage. We are also thankful for the reviewer's hints on the occurrences of continuous glass transitions on other systems, which gave us the opportunity to draw broad analogies with our own. We made extensive changes and additions in the text at two places. First, at the Results section, we added the following passage:

A striking characteristic of the ensuing arrested state is that it features a single, β -relaxation process and the subsequent α -relaxation is absent, in contrast to the common, two-step relaxation scenario [33] encountered for polymer glasses [34] or for repulsive colloids [35]. Indeed, we have not been able to observe the α -relaxation, despite the fact that we have extended our simulations to very long times, over $22\tau_{\text{diff}}$. The absence of a two-step process is a feature associated with continuous, type-A glass-transitions, as opposed to the discontinuous, type-B transitions [36-38], and it implies the presence of higher-order singularities, the so-called A_3 - and A_4 -critical points, in the framework of Mode Coupling Theory [39-42]. There are strong indications that the system at hand features such higher-order singularities, a point to which we will return in the Discussion section.

Further, we introduced a Discussion section, in which we entered a more detailed discussion on the topic as follows:

We now return to the question of the order (continuous or type-A vs. discontinuous or type-B) of the glass transition for the system at hand. The usual control parameters driving vitrification in molecular or colloidal systems are the temperature and the density, and the typical glass transition scenario there is discontinuous: the intermediate scattering function in the ergodic state develops a plateau, which grows in height and extends longer in time approaching the glass transition as the control parameters are changed [10,33,50]. In the continuous case, the non-ergodicity factor grows smoothly from zero to finite values. This second scenario is less common and its realization requires the presence of additional control parameters, such as porosity and randomness [36-38], tunable attraction widths in the interactions [51,52] or confining periodic potentials [53] for which the amplitude and the wavelength can be independently varied. In such cases, lines of continuous glass transitions have been found in parameter space, and they are associated with higher-order singularities of the A_3 - or of the A_4 -type. The latter appear as endpoints of type-A transition lines that merge with type-B lines, as endpoints of type-B lines separating two glasses or as endpoints of A_3 -lines in the latter case.

For the system at hand, several control parameters can be tuned: (i) the fraction M_a/M of partly-active rings; (ii) the ratio T_h/T_c of the temperatures of the hot and cold segments; (iii) the fraction N_h/N of hot segments on a ring; and (iv) the number of monomers N of the rings. The richness of the system makes the possibility of existence of higher-order singularities in principle possible. A detailed investigation in the vast space spanned

by these is beyond the scope of this work; we focused mainly on cases (i) and (ii) above. In Figures 8a and 8b, we show the effect of gradually increasing the fraction of partially active chains, which induces a glassy state, as witnessed by the saturation of the mean-square displacement, Figure 8a, and the growth of a non-ergodic plateau, Figure 8b, as the ratio M_a/M exceeds a number as small as 1/160. There is no evidence of the development of an intermediate plateau in the ergodic state preceding the glass, in full analogy with type-A transitions seen in the aforementioned systems [36-37,51-53]. The presence of a subdiffusive regime in the mean-square displacements, Figure 8a, and of a logarithmic crossover intermediate scattering function, Figure 8b, offer additional corroboration that the transition for this choice of the remaining system parameters is continuous, and thus higher-order singularities are present. We have found similar behaviour (not shown) varying the ratio T_h/T_c . Naturally, this does not rule out that in other parts of the phase diagram the transition is governed by A_2 -singularities and it is thus of B-type. This would give rise to a number of additional scenarios for the behaviour of the relaxation functions, including the possibility of multiple relaxations observed in related models [53,54]. The presence of deep and tight threadings in our system bears an intuitive analogy with colloidal attractions [52], random pinning [36-38,50] or polymer-mediated bond formation [54,55] in systems featuring similar glass-transition phenomenologies. A detailed investigation of this issue, however, is a problem for the future.

- (ii) *Their results clearly show there is aging in the system. This is not surprising since the protocol the authors are following should naturally give rise to aging. The quench in this system is in activity and not temperature or density [Phys. Rev. Lett. 109, 115702 (2012)]. This is interesting from the glass perspective as well. In my opinion, the authors should refrain from making comments like "... t_0 plays the role of the volume fraction ...". One can probably view the system in terms of an evolving effective volume fraction, but I think this is oversimplifying the main perspective of aging.*

Reply: The reviewer is quite correct. We have now acknowledged this point and we removed this statement from the revised version.

- (iii) *In an aging system, key characteristics of the system changes with the aging time t_0 . It will be important to show a more quantitative characterization of this aspect. The authors could show how relaxation time and the non-ergodicity parameter, $f(q, t_0)$, change with t_0 . Then consider two different quenches, for example for two different number of active rings, and show if these functional dependences remain same. Do the exponents change from those of a passive system?*

Reply: The relaxation time τ_α in the context of glassy dynamics is usually defined as the time for which the intermediate scattering function (self- or collective one) attains a value equal to half of the plateau value during the α -relaxation; this is then determined for different waiting times t_0 and plotted against the latter, typically featuring a power-law dependence on it. Since we do not have/we do not see the α relaxation here, we opt for defining, in analogy, a time scale τ_β as the time for which the intermediate scattering function attains a value $(1 + F_{s^*})/2$, where F_{s^*} equals the plateau value of the corresponding ISF. Results are shown in the new Figure S11b, and a power-law dependence of τ_β on τ_0 can be seen, albeit with an exponent $\cong 0.2$, much smaller than the values seen for colloidal glasses, which are $\gtrsim 1$ [A. M. Puertas, M. Fuchs and M. E. Cates, J. Phys.: Condens. Matter **19**, 205140 (2007)] in the case of α -relaxation.

- (4) *Fig. S2(d): Could the authors show it in logarithmic scale? I would expect a similar behavior as $F_s(q, t, t_0)$. The reason is related to some of the theoretical approaches and not particular to the current system; in the glassy regime, relaxation of any two-point quantities should show similar behavior. Of course this is an approximation, but it will be nice to see if that holds in this system as well.*

Reply: According to the suggestion of the reviewer, we have replotted Fig. S2(d) semi-logarithmically in time and it indeed features similar relaxation behavior as the self-part of the intermediate scattering function $F_s(q, t, t_0)$ for a given waiting time t_0 . Namely, for all t_0 we observe a first β -like relaxation stage lasting for approximately $10^5\tau$ followed by a plateau. Such β -relaxation times are similar to that of $F_s(q, t, t_0)$, as seen in Fig. 3(a). Moreover, for waiting times $t_0 > 10^5\tau$ the directional relaxation function $\langle \mathbf{e}_{\text{ch}}(t + t_0) \cdot \mathbf{e}_{\text{ch}}(t_0) \rangle$ seems to start relaxing further off the plateau for $t > 5 \cdot 10^6\tau$, however, as in the case of F_s , we do not observe a full relaxation. The fact that the correlation function $\langle \mathbf{e}_{\text{ch}}(t + t_0) \cdot \mathbf{e}_{\text{ch}}(t_0) \rangle$ seemingly relaxes off the plateau might imply that the α -relaxation of the intermediate scattering functions also occurs in the system, but possibly at much larger timescales, which are at present inaccessible computationally.

REPLY TO REVIEWER 2

In this work Smrek and colleagues build upon their recent work on dense solutions of linear polymers with different temperatures and this time consider ring polymers. This seemingly small change has a big impact on the dynamics of the system and indeed they provide computational evidence for the onset of an "active topological glass" triggered by activity. The extensively characterise this state and, finally, they draw a connection with the behaviour of chromatin in vivo.

I think that this is a very fundamental piece of work in several respects and I strongly support its publication in Nature Communication. I believe it will be considered seminal in both polymer physics and active matter (and perhaps even in genome organisation) communities.

Reply: We thank the reviewer that s/he has recognized our work as fundamental and worthy publication.

At the same time, I also feel that some sections may be improved or made more precise as follows:

1. *"The rings cannot cross each other, but are known to thread — one ring piercing through an eye of another ring (see Fig. 5a), which temporarily topologically constraints the motion of the two rings". I would argue that it is only the passively threaded ring whose motion is constrained.*

Reply: Although such asymmetric effect of the constraints has been proposed and its impact on the dynamics investigated on a simplified model [E. Lee, Y. Jung, *Polymers* 11(3), 516 (2019)], we respectfully disagree. While the motion of the actively threading ring is not constrained in the longitudinal motion of the threading direction, the transversal motion is restricted by the presence of the passively threaded ring. As the rings do not relax by reptation [Halverson et al *J. Chem. Phys.* 134, 204905 (2011).], we should consider relevant both constraint directions. Moreover, the threading direction is likely relevant only on the local scale and due to the highly crumpled conformation of the rings, the threadings are frequently symmetric - two rings tend to actively thread each other [J. Smrek, A. Y. Grosberg, *ACS Macro Lett.* 5, 750–754 (2016): Figure S6].

We think that the details of the threading constraints are not yet fully understood in equilibrium and even less so in the present system. Therefore we rather restrained from statements on the specifics of the threading constraints. As we point out in the conclusion it is an open question to uncover the threading topological order parameter that governs the glass transition. We believe that answering this question in future for the present active topological glass can also help us to understand under what condition can one expect to find the equilibrium topological glass.

2. *"... which would give rise to exponentially long relaxation times". If there is a critical parameter for which vitrification occurs, then the rise of relaxation time is expected to be super-exponential (or exponential with divergence, see VFT law for glass-forming systems) which is seen in the case of pinning. Otherwise, if it is a cascade of relaxations then one may expect just a steep power-law (?). With this comment, I just want to point out that it is not clear (at least to me) that the relaxation due to hierarchical threadings necessarily leads to an exponentially long relaxation. Can the authors comment? Maybe it would be interesting to quantify the relaxation time of the rings in their simulations by measuring the effective diffusion coefficient against the ratio of hot/cold temperatures. I understand that this would require a whole set of new simulations and since there are already enough results in this paper perhaps the authors may want to consider this suggestion for the future.*

Reply: We agree with the reviewer that the specific functional dependence for the relaxation time of the hierarchical threadings in equilibrium has not been completely clarified. We used "exponentially" as the exponential relaxation with the polymer length N has been established in the simplified model of topological glass [W. Lo, M. S. Turner, *EPL* 102, 58005 (2013)]. Although that model simplifies the ring conformation in melt, the exponential arises due to the requirement of the sequential relaxation of threadings, the number of which grows with N . These requirements are plausible for the equilibrium melt of rings as well, e.g. the growth has been observed in equilibrium simulations [J. Smrek, A. Y. Grosberg, *ACS Macro Lett.* 5, 750–754 (2016)]. However, we acknowledge that it is unknown if the number of threading neighbors in equilibrium, in high N limit, is sufficient for the effect to take place and what is the effect of the realistic ring conformation on the hierarchy of threadings. Therefore, we adjusted the statement in the introduction accordingly:

... which could give rise to very long relaxation times e.g. exponential in the ring length [6]

We agree the dependence of the diffusion properties on the temperature ratio is of high importance and we plan to investigate it in future in the context of also other control parameters that shape the phase diagram.

3. *Just to iterate on the point above. The authors mention arm retraction in page 7 and argue that the relaxation of stars and unthreading are similar in nature. I disagree with this in the sense that different arms of a star are*

independent, but any section of a ring contour is directly connected to any other. Rings lack branching points which decouple the relaxation of different segments. So even if a section of a ring is threading a neighbour, then the diffusive motion of the *non-threading* section affects the other and speeds up the unthreading by "dragging" it away. Perhaps the authors can comment on this?

Reply: We meant the arm-retraction only as an analogy for the unthreading event, but we agree with the reviewer that the analogy might not be accurate enough. We rephrased the sentence accordingly:

However, the directionality and the presence of other rings oppose the back-tracking.

4. "Then, in the limit of long rings and high density a glassy behavior can be extrapolated to $f \rightarrow 0$ ". Actually it is the other way around: the extrapolation (done using any range of chain length and density) to $f \rightarrow 0$ is done to find the critical values (of length and density) for which pinning is no longer required.

Reply: We agree with the reviewer and we apologize for the confusing wording. We meant that the topological glass without pinning can only appear above certain ring length and density. We have reformulated the sentence accordingly:

Then, for sufficiently long rings and high density a glassy behavior can be extrapolated to $f \rightarrow 0$

5. "...it is unlikely to be utilized experimentally to drive the topological glass transition". This is a strong statement which is unnecessary and unsupported. For sure, the proposed activity creates a completely different (and perhaps better) route to achieve topological freezing but one can still envisage a number of (experimentally challenging) set-ups in which a fraction of the rings is slower than the other and thus impose an effective pinning...

Reply: We agree with the reviewer and we have softened the statement:

... creating it experimentally to drive the topological glass transition would be challenging

6. The authors state that the temperature for the hot regions is 3 times larger than that for the cold ones. This can hardly be considered "modest" to my opinion. What is the rationale for choosing this number (apart from driving the microphase separation which in any case is unrelated to the glassy dynamics)? It also begs the question of how can one realise such a strong difference experimentally... I think it would be useful if the authors could perhaps add a sentence or two on this (experimental realisations) in the conclusions.

Reply: Indeed the temperature contrast 3 is relatively high and we have chosen it to speed up the overall dynamics. We plan to investigate the phase diagram as function of the temperature ratio as well. Preliminary results confirm that indeed the glass transition occurs also at significantly lower temperature ratios, but the onset takes longer. We mention this fact in the revised manuscript in the Conclusions and Methods section.

For the experimental realization we added some more details in the conclusions:

The effective temperature ratio T_h/T_c of a factor of three that we used here is within reach, since ATP hydrolysis releases more than $10k_B T$ [84]. However, as our preliminary results suggest, even smaller temperature ratios can be sufficient. Other means of selective heating could be attempted by fluctuating external fields or IR irradiation selectively coupling to individual groups in the polymer [85].

7. By looking at fig.5c I get the impression that, effectively, turning on the activity creates a bias towards creating threadings rather than removing them; in other words, the rates of threading/unthreading are no longer equal (as they are expected to be in equilibrium for detailed balance) and are skewed by the activity. My question to the authors is: do you think the long-time value you observe is because the system at that point is glassy and cannot create new threadings anymore or is it a new steady state? i.e., do threadings still exchange?

Reply: This is an excellent question and we have investigated it in more detail. As the reviewer correctly pointed out the equilibrium balance is shifted towards the threading creation at early times. At late times (glassy regime) however, the system arrives again at a steady state when the creation and annihilation of threadings is non-zero, but balanced at a higher total threading level in comparison to equilibrium. Interestingly, not all threadings are being intermittent. In the glassy regime we measured the distribution of threading survival time and find it to be bimodal. A set of threadings are relatively short-lived $< 10^5 \tau$, while others are persistent, appearing early after the activity onset and remain for the whole simulation duration. We have included these new findings in the revised manuscript and the Supplementary Material:

While the cluster profile remains stable in the glassy regime, we still observe the threading and un-threading events between ring pairs. Numbers of these events, however, balance each other and we observe a steady state (Fig. 6a). Interestingly, the threading life-time distribution shows a power-law with a peak at late times,

meaning, the majority of threadings being persistent and surviving for times longer than the simulation duration and a minority having a short lifetime. The bimodal character is likely a consequence of the fat power-law tail as revealed by the shape of the distribution measured at different times after the activity onset (Fig. 6b and Supplementary Video 2). Additionally, the threading two-point correlation function $\Phi(t, t_0)$ (defined in Methods) exhibits incomplete relaxation at all times showing dynamic threading steady state with persistent threadings (Fig. 6a).

We could not determine the class of the persistent threadings but we plan to investigate their detailed topology in future.

- 7a. *Iterating on 7, do you think that if it was possible to simulate (or realise) rings that display about 20 threadings per ring *in equilibrium*, one should expect glassy dynamics without either pinning nor activity? In other words, is 20 threadings another "magic number"? I am saying this because 20 is also the "magic" Kavassalis–Noolandi number to which the overlap parameter converges to for large rings (see Rosa, Everaers PRL 2014 and Ge et al Macromol 2016). I wonder if this is only a coincidence or if one needs, in practice, to thread all neighbours in order to vitrify.*

Reply: The relevance of the 20 threadings per ring is an interesting idea for the equilibrium, but it certainly does not apply to our present system. The system consisting of rings of length $N = 200$ is glassy, but there are only about 9 threadings per ring as was shown in Fig. S11g.

We agree that the number of threadings could matter in equilibrium if they are sufficiently deep. For example, if only threadings deeper than the entanglement length are counted, their mean number and their distribution as well are independent of the polymer model, as shown in [J. Smrek, K. Kremer, A. Rosa, ACS Macro Lett. 8, 155–160 (2019)]. When the shallow threadings are discarded the currently longest equilibrated systems ($N/N_e = 57$) show only about 7 threadings per ring (see Fig. S4 of that paper). This number grows with the ring length and can be extrapolated to reach the value of 20 somewhere around $N/N_e = 200$. This is above the predicted onset ($N/N_e \simeq 90$) of equilibrium topological glass from the pinning works [D. Michieletto, N. Nahali, A. Rosa, Phys. Rev. Lett. 119, 197801 (2017)].

In the present work we counted threadings of any length, because (i) we observe even short tight threadings (that are entropically unfavourable in equilibrium) that might be relevant for the system's stability and (ii) it is not clear if the notion of the entanglement length can be easily carried over to the present active system.

- 7b. *Iterating on 7 and 7a, can the authors measure and report the average number of threadings per chain in the systems with varying number of active chains (Fig. 7)? Do the systems need to reach $\langle n_{tn} \rangle = 20$ in order to freeze?*

Reply: We report this in the new Fig. 8c. The average number of threadings per chain grows with the number of active chains in the system, but it does not necessarily reach the value of 20 although the system is already in the glassy state. Iterating on the reply to 7a, the number of threadings itself is not an universal indicator of the transition. Additionally, we observe the quantity $\langle Q \rangle$ (Fig. S9) also grows with the number of active chains in the system, which shows the threadings are on average deeper when more active chains are involved. This is in line with our reported observation that the threading clusters are supported by the active chains, which due to their larger extent thread more than the passive ones.

While all the evidence shows the correlation of the number and the depth of the threadings with the onset of the glassiness, we do not put these forward as proper order parameter for the transition. We have stated this in the context of persistent threadings and in the conclusions. We plan to investigate it further in future.

These threadings are likely members of the persistent class of threadings, but we could not determine the latter class yet.

On the biology side:

8. *I have never been a fan of the mapping ring polymers \rightarrow chromatin in vivo. I think it effectively works well, but we still do not understand why. In particular there is one strong argument against it: the fact that the topological constraints in the two systems are completely different. In vivo, our chromosomes are not only not rings (some relatively small sections of them may topologically look like rings, e.g. looped TADs) but are also constantly cut and glued back by Topoisomerases. The latter entails that there is no global topological invariant that is preserved, oppositely to standard melts of rings which are simulated here and elsewhere.*

Reply: We are aware of the complications of modeling of chromosomes as rings. For completeness, the model of rings with permanent topological constraints is a simplification for the constraints that may arise due to insufficient equilibration time (or energy) and the lack of reptation relaxation mechanism (see details in [J. D. Halverson et al, Rep. Prog. Phys 77, 022601 (2014)]). The latter is supported by the existence of the association of some segments with the nuclear lamina e.g. [M. Falk et al Nature 570, 395–399 (2019)]. Indeed the chromosomes do have ends and the topoisomerase is present in the nucleus. However, to our current understanding it is not clear if the topo II (energy-dependent) activity is extensive enough to make the chromatin fiber completely phantom for the topological constraints to vanish on all length and time scales. If it was so, there would have to be another mechanism for keeping the chromosomes (territories) unmixed as well as the TADs unlinked. While alternative, organization models, such as “strings and binders switch” or copolymer models consider chains equilibrated, therefore implicitly crossable, they concentrate on finer structures and as such have open questions too, such as: (i) do the binders have to be chromosome specific to prevent the mixing?; or (ii) does the interaction need to be fine-tuned for different cell conditions?

Unable to explain the organizational conundrum at all scales satisfactorily at this point, we assume the existence of the topological constraints at least for the typical observation times and examine the consequences when also activity is present in the system. In this line, we intentionally do not insist on whether the partly active rings represent the chromatin on TAD level or large scale structures, but rather present a range of phenomena where it could play a role, given the assumptions.

To put in context the hypothesis of relevance of the current work to chromatin, we mention these issues briefly in the revised manuscript and cite appropriate references for a more complete overview. To see the changes, please see the manuscript or the text in red after reply to the question 10.

9. *I acknowledge that the effectively "hot" regions in the authors' simulations are good representations of transcribed genes (and other active sections of the genome) *but* the segregation of active/inactive compartments and heterogeneous and subdiffusive (or glassy) dynamics can also be explained by (and perhaps more simply) copolymer models (Jost et al NAR 2014, Michieletto et al PRX 2016), bridge/binding proteins (Shi et al, Nat Comm 2018) or even just confinement (Kang et al PRL 2015). It is unclear, from the way this section is written, if the authors imply that modellers should model the genome as a ring or if they should include regions with differential activity (or both).*

Reply: We intentionally do not claim that the activity-driven phase separation and the glassy dynamics arising from the topological constraints under active driving are the main governing mechanisms of the observed static and dynamic properties of the chromatin in vivo. Indeed, the euchromatin and heterochromatin are chemically different and therefore their equilibrium phase separation would be difficult to avoid owing to their polymeric nature. However, in our opinion it is important to account for all physical phenomena that can potentially play a role, explain their mechanisms and then judge their respective contributions. Here we report on a single mechanism of the interplay of the activity and topological constraints, which is responsible for a range of non-equilibrium effects that are in line with some observations of chromatin at various length and time scales. Whether nature has taken this “opportunity” is in our opinion premature to say. Therefore, we do not suggest what the genome modellers should do in general, but rather we underline in which context and under what conditions these effects could potentially play a role and should be taken into consideration. Certainly more work is necessary to understand the active phenomena we report here in more detail.

We agree that the alternative mechanisms should be mentioned in more detail and we have rewritten this section accordingly. Please see the section in the manuscript or here after our reply to the question 10.

10. *Finally, in this paper the authors find glassy behaviour due to threadings, but these constraints are easily resolved by TopoII in vivo. So perhaps the origin of slow dynamics of chromatin must be found elsewhere rather than threadings (see point 9).*

With these comments I don't intend to put off the authors from drawing this connection which is very fascinating, but perhaps they should be more careful with their wording and try to give a more complete and accurate picture (albeit this is far from complete yet and so not an easy task!).

Reply: We have commented on the role of the TopoII in the reply to question 8. We agree that a more complete picture would be appropriate and we attempt to give it in the revised version of the section in the manuscript:

Connection to chromatin conformation and dynamics

While our model system is interesting from the fundamental physics point of view, it is also inspired from biology and may well bear important ramifications for the organization of the DNA fiber (chromatin) of higher eukaryotic cells. The large-scale static properties of the equilibrium melt of unknotted rings, such as the territorial organization, the scaling of $\langle R_g \rangle$ with N or the so-called contact probability (Supplementary Fig. S1d) are consistent with the population-average conformation of the interphase chromatin [57]. This can be due to the common governing role of the topological constraints in both systems [57]. The rationale behind the ring model is the time scale separation - the chromosomes are linear chains and as such equilibrate and tangle by reptation, however for the length and density of chromatin, such relaxation would take significantly longer than the cell's life time [58]. Therefore the constraints arising from the uncrossability of the chains can be modeled as permanent on biological time scales. This is done effectively by the closure of the chain's ends which inhibits reptation (see also [59] for other view on the relaxation in this context). Furthermore the chromatin association with the nuclear lamina [60] hinders reptation relaxation and the chromatin can be viewed as loops between lamina contact points. In contrast, the topoisomerase II enzyme can resolve the topological constraints, by cutting the fiber, passing segment through sealing up the cut back [61], although the extent to which it affects global topology on biological time scales in interphase is unclear. Alternatively, the rings can also represent small scale chromatin loops extruded or maintained by Structural Maintenance of Chromosomes protein complexes [62,63] and/or Topologically Associating Domains [64] that do not link. Current experimental evidence for the topological state (knottedness) of the chromatin fiber also varies. While conformations inferred from contact probability measurements exhibit knots [65], the knot analysis [66] of the direct observations of fluorescently labeled chromatin segments finds mostly unknotted segments. Unable to refine the scales of the topological constraints completely at this point, we assumed their existence for the typical observation times and examined the consequences. Therefore, we mention phenomena on chromatin at various scales for which the interplay of the topology and the activity could be relevant.

The segmental activity with thermal-like fluctuation spectra has been measured to be stronger in the normal living cell nuclei [67] as opposed to energy-deprived cells, and has its origins in the energy dependent processes, such as chromatin repair or remodeling, DNA transcription, or loop extrusion. We conjecture that the phenomena observed in our partly-active system could also be relevant in biological context, on the basis of the following similarities with our model: *(i)* genes exhibit size increase upon transcription decoupled from the chromatin decondensation [68], *(ii)* a highly transcribed gene shows directed motion [69], *(iii)* overall chromatin loci exhibit heterogenous subdiffusive dynamics [76] reminiscent of glassy behavior [24,70] *(iv)* the active and inactive chromatin are spatially segregated [71,72], and *(v)* the chromatin exhibits a doubly-folded structure at small scales [73]. Naturally, these effects have also alternative explanations. The phase separation and glassy dynamics could be observed for a copolymer models where different chromatin segments have different interaction potentials based on their epigenetic state [24, 70, 74, 75] or by interaction with binding proteins [75,76]. The double folded structure is likely to be attributed to supercoiling due to the torsional stress induced on the fiber in the process of transcription or loop extrusion [62,77–80]. Nevertheless, we show that the activity in combination with topological constraints at the fiber level can complement the observed phenomena and should be considered in a more complete picture of the chromatin organisation [81,82]. As the models above typically do not consider topological constraints and activity, our findings represent a completely new mechanism for the observed phenomena. Certainly, more work is required to determine the relative contributions of the different mechanisms in the various cases. Potentially, the different nature of the observed glassy states, namely the monomer glass due to caging in copolymer models [24,70] and topological glass here, could be used to discern which one could be present or dominant in the chromatin context.

REVIEWERS' COMMENTS:

Reviewer #1 (Remarks to the Author):

I have gone through the revised manuscript and the response to the reviewer comments/suggestions. I feel the authors have adequately responded to the suggestions and satisfactorily answered the questions raised by both the referees. This is an important piece of work, although the connection to the chromatin dynamics remains somewhat speculative, I believe, the current work should inspire deeper exploration of the problem.

I do not have any additional comment on the manuscript and strongly recommend acceptance.

Saroj Kumar Nandi

Reviewer #2 (Remarks to the Author):

The authors have done a great job at addressing my comments.
I am fully supportive of this revised version in Nature Communications.